# Higher-order interactions enhance the latitudinal tree diversity gradient

Yuanzhi Li[1,51], Junli Xiao[1,51], Yuan Jiang[1], Stuart Joseph Wright[2], Margaret M. Mayfield[3], Oscar Godoy[4], Alfonso Alonso[5], Kristina J. Anderson-Teixeira[2,6], Jennifer Baltzer[7], Joseph D. Birch[8], Pulchérie Bissiengou[9], Norman A. Bourg[6], Warren Brockelman[10], David F. R. P. Burslem[11], Min Cao[12], Keith Clay[13], Stuart J. Davies[14], Qingqing Du[15], Sisira Ediriweera[16], Anna Feistner[17], Edwino S. Fernando[18], Gregory S. Gilbert[19], Zhanqing Hao[20], Jan Holík[21], Mingxi Jiang[22], Guangze Jin[23], Daniel J. Johnson[24], Alexander S. Jones[25], Kamil Král[21], Andrew J. Larson[26], Buhang Li[27], Juyu Lian[28], Luxiang Lin[12], Feng Liu[29], Yu Liu[30], Zhili Liu[23], James A. Lutz[31], Keping Ma[32], Sean M. McMahon[2,33], William McShea[6], Hervé Roland Memiaghe[34,35], Xiangcheng Mi[32], Jonathan A. Myers[36], Musalmah Nasardin[37], Anuttara Nathalang[10], Michael J. O'Brien[38], Nestor Laurier Engone Obiang[34,39], Geoffrey Parker[33], Richard P. Phillips[40], Xiujuan Qiao[22], Haibao Ren[32], Glen Reynolds[41], Lillian Jennifer V. Rodriguez[42], Pavel Šamonil[21], Guochun Shen[43], Zufei Shu[44], Jessica Shue[2,33], Mark E. Swanson[45], Jill Thompson[46], María Uriarte[47], Xihua Wang[43], Xugao Wang[48], Youshi Wang[27], Tze Leong Yao[37], Wanhui Ye[28], Mingjian Yu[49], Minhua Zhang[30], Yan Zhu[15], Jess Zimmerman[50], Fangliang He[8,30] & Chengjin Chu[1✉]

The global decrease in species diversity from low to high latitudes is among the most robust biogeographic patterns[1,2]. There is continuing debate on the contribution of conspecific negative density dependence (CNDD) to the latitudinal diversity gradient evident for trees[3,4]. Theory suggests that CNDD based on pairwise interactions alone is not sufficient to explain the intricacies of diverse communities, because higher-order interactions (HOIs) may greatly modify these interactions[5,6]. However, there has been a lack of empirical studies investigating how HOIs intertwine with pairwise interactions and how they may contribute to the latitudinal tree diversity gradient. Here we examined both pairwise interactions and HOIs across 32 large permanent forest plots, most in the northern hemisphere. We detected evidence of HOIs in 40% of the 1,543 species–plot combinations for tree growth, and 23% of the 1,340 such combinations for tree survival, with the strength of these interactions declining with latitude. HOIs were found to benefit rare species but disadvantage common species, suggesting a potential mechanism promoting species diversity. This stabilizing effect weakened towards higher latitudes, consistent with the latitudinal tree diversity gradient. Our findings reveal an important interplay between pairwise interactions and HOIs in promoting the latitudinal tree diversity gradient and help to clarify the contribution of CNDD to this biogeographic pattern.

The systematic decline in species diversity from the equator to the poles, known as the latitudinal diversity gradient, is among the most widely observed biogeographic patterns[1,7]. The latitudinal decline in tree species diversity is particularly prominent and well documented[2,8]. CNDD, whereby conspecific neighbours exert more negative effects on the performance of a focal tree than heterospecific neighbours[9], is considered to be a primary mechanism that maintains local diversity[10] and promotes the latitudinal tree diversity gradient[11,12]. The lemma that CNDD regulates latitudinal diversity gradient posits that CNDD is stronger in tropical than in temperate forests[4]. However, empirical evidence for this proposition is mixed: some studies based

on field data across forest plots have reported significant declines in CNDD with increasing latitude[12] or decreasing species richness[11], whereas others did not support this trend[3,13,14]. These inconsistent findings have led to a longstanding debate on whether there is a latitudinal pattern of CNDD and whether it contributes to latitudinal tree diversity gradient[4,15–18].

Although negative density dependence has been the main focus of interest in previous studies of the latitudinal tree diversity gradient, positive density dependence (facilitation) generated by mutualists (for instance, mycorrhizal fungi)[19] could also play an important part in maintaining local species diversity[20,21] and contribute to the

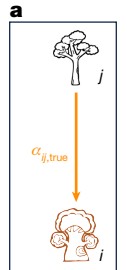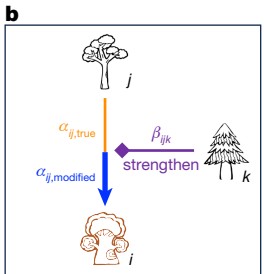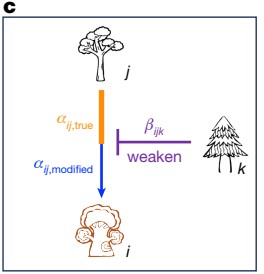

**Fig. 1 | Effect of a neighbouring tree of species *j* on a focal tree of species *i* in the absence and presence of another neighbouring tree of species *k*. a**, The pairwise effect of a neighbour of species *j* on the focal tree of species *i* in the absence of other neighbours is denoted $\alpha_{ij,\text{true}}$. **b,c**, In the presence of another neighbour of species *k*, the higher-order effect of *k* (initiator) on *i* (receiver) through *j* (transmitter), denoted $\beta_{ijk}$, could either strengthen (**b**) or weaken (**c**) $\alpha_{ij,\text{true}}$.

latitudinal diversity gradient[22,23]. Moreover, existing studies have mostly been built on the assumption that organisms interact in a pairwise fashion[5,24], neglecting the potential effects of HOIs that emerge when pairwise interactions between two neighbouring trees are modified by other neighbours (Fig. 1). Specifically, the effect of a neighbouring tree of species *j* on a focal tree of species *i* in the absence of other trees, denoted $\alpha_{ij,\text{true}}$ (Fig. 1a), can be competitive ($\alpha_{ij,\text{true}} < 0$) or facilitative ($\alpha_{ij,\text{true}} > 0$), and conspecific ($j = i$) or heterospecific ($j \neq i$). This pairwise effect can be either strengthened (Fig. 1b) or weakened (Fig. 1c) in the presence of another neighbouring tree of species *k*. The higher-order effect of *k* (initiator) on species *i* (receiver) through *j* (transmitter) is denoted $\beta_{ijk}$. As a consequence, the effect of *j* on *i* in the presence of *k*, denoted $\alpha_{ij,\text{modified}}$ (implicitly incorporating $\beta_{ijk}$), can be stronger (Fig. 1b) or weaker (Fig. 1c) than $\alpha_{ij,\text{true}}$, depending on whether $\alpha_{ij,\text{true}}$ and $\beta_{ijk}$ have the same or opposite signs. As an example, *Eucalyptus urophylla* (initiator) can inhibit the root growth of *Cryptocarya concinna* (transmitter) through allelopathic effects[25], thereby weakening the competitive effect of *C. concinna* on its neighbouring species (receiver).

There is accumulating evidence that HOIs regulate the survival and growth of trees[26,27] and thus play a substantial part in the structuring of community assemblages[28–30]. Consequently, estimating CNDD (when $\alpha_{ii} < \alpha_{ij} < 0$) without explicitly considering HOIs may evoke divergent latitudinal changes in CNDD and obscure the true contribution of CNDD to the latitudinal tree diversity gradient. To better understand the contributions of pairwise interactions and HOIs, we examined how their cumulative effects changed with species abundance and latitude. Pairwise interactions and HOIs may act as stabilizing forces promoting species diversity when their cumulative effects are negatively associated with species abundance (that is, rare species are less negatively or more positively affected). Such stabilizing effects may further reinforce the latitudinal tree diversity gradient if this cumulative effect–abundance relationship weakens with increasing latitude. It has remained unclear, however, whether HOIs are common in forests and how they contribute to the latitudinal tree diversity gradient.

In this study, we assembled census data from 32 large permanent forest plots spanning tropical to boreal forests (Fig. 2a and Supplementary Table 1) to address three key questions. (Q1) Are HOIs prevalent among trees across forest plots? (Q2) How do pairwise interactions ($\alpha_{ij,\text{true}}$ and $\alpha_{ij,\text{modified}}$) and HOIs ($\beta_{ijk}$) vary with latitude? (Q3) How do latitudinal changes in HOIs contribute to the latitudinal tree diversity gradient (Fig. 2b)? To answer these questions, we estimated both pairwise interactions and HOIs from demographic growth (for 1,543 tree species–plot combinations) and survival models (for 1,340 tree species–plot combinations). With these data, we built three types of growth

and survival model: (1) null models with no biotic interactions, (2) pair-only models including only pairwise interactions and (3) HOI-inclusive models including both pairwise interactions and HOIs. The pairwise interactions estimated from pair-only models implicitly incorporate the effects of HOIs and thus represent $\alpha_{ij,\text{modified}}$ (blue arrows in Fig. 1), whereas pairwise interactions estimated from HOI-inclusive models isolate HOIs and thus represent $\alpha_{ij,\text{true}}$ (orange arrows in Fig. 1), the true interaction between two trees of species *i* and *j* in the absence of other neighbours. We then compared Akaike information criterion (AIC) values of the three types of model (Q1), tested whether the estimated pairwise interactions ($\alpha_{ij,\text{true}}$ and $\alpha_{ij,\text{modified}}$) and HOIs ($\beta_{ijk}$) declined with latitude (Q2), and evaluated how the cumulative effects of pairwise interactions and HOIs on growth and survival changed with species abundance and latitude (Q3).

## Prevalence of HOIs across forest plots

Our results demonstrate strong statistical support for the prevalence of HOIs across 32 large permanent forest plots, as evidenced by the AIC values of HOI-inclusive growth models being at least two units lower than those of the alternative models for 40% of the 1,543 species–plot combinations across all 32 forest plots (Fig. 3a). Moreover, HOI-inclusive growth models significantly improved predictions of tree growth rates compared with the alternative models (Fig. 3b). Although HOIs have been reported at local scales in forests[26,27] and other systems (for instance, bacteria–paramecium–protozoan[7], amphibians[31], microcrustaceans[32], annual plants[33]), our study shows that HOIs are ubiquitous in global forests. The widespread HOIs in forests highlight the importance and necessity of further exploration of their latitudinal patterns and ecological consequences.

## Latitudinal decline in biotic interactions

Competitive ($\alpha < 0, \beta < 0$) and facilitative ($\alpha > 0, \beta > 0$) interactions were almost equally frequent across all the species–plot combinations (Extended Data Fig. 1). The average strength of true intraspecific pairwise interactions ($\alpha_{ii,\text{true}}$), both competitive and facilitative, declined rapidly with latitude (orange dots and lines in Fig. 4a). However, the average strength of true interspecific pairwise interactions ($\alpha_{ih,\text{true}}$), both competitive and facilitative, remains relatively constant across latitudes (orange dots and lines in Fig. 4b). These results are consistent with some previous work indicating that higher tree diversity is associated with stronger CNDD[11,12]. HOI coefficients (except $\beta_{ihh}$), both competitive and facilitative, also declined with latitude (Fig. 4c–f).

Strong negative correlations were detected between $\alpha_{ii,\text{true}}$ and $\beta_{iih}$ ($r = -0.96$) and between $\alpha_{ih,\text{true}}$ and $\beta_{ihh}$ ($r = -0.89$) (Extended Data Fig. 2). This meant that both intraspecific ($\alpha_{ii}$) and interspecific ($\alpha_{ih}$) pairwise interactions were more strongly weakened by heterospecific neighbours ($\beta_{iih}: \alpha_{ii} \leftarrow h$ and $\beta_{ihh}: \alpha_{ih} \leftarrow h$) than by conspecific neighbours. Consequently, the latitudinal signal of intraspecific pairwise interactions modified by HOIs ($\alpha_{ii,\text{modified}}$) became much weaker (blue dots and lines in Fig. 4a) or even disappeared (Supplementary Text 2.3), because $\beta_{iih}$ weakened $\alpha_{ii,\text{true}}$ more strongly in low latitudes than in high latitudes (Fig. 4e). By contrast, interspecific pairwise interactions ($\alpha_{ih,\text{true}}$) were weakened uniformly by relatively constant $\beta_{ihh}$ across latitudes (Fig. 4f), resulting in relatively constant modified interspecific pairwise interactions ($\alpha_{ih,\text{modified}}$) (blue dots and lines in Fig. 4b). Our findings suggest that HOIs, by modulating pairwise interactions across latitudes, may reconcile the divergent observations of latitudinal changes in CNDD[3,11,12].

## HOIs enhance latitudinal tree diversity gradient

Previous studies have primarily focused on competitive pairwise interactions (that is, CNDD) and their contribution to the latitudinal tree

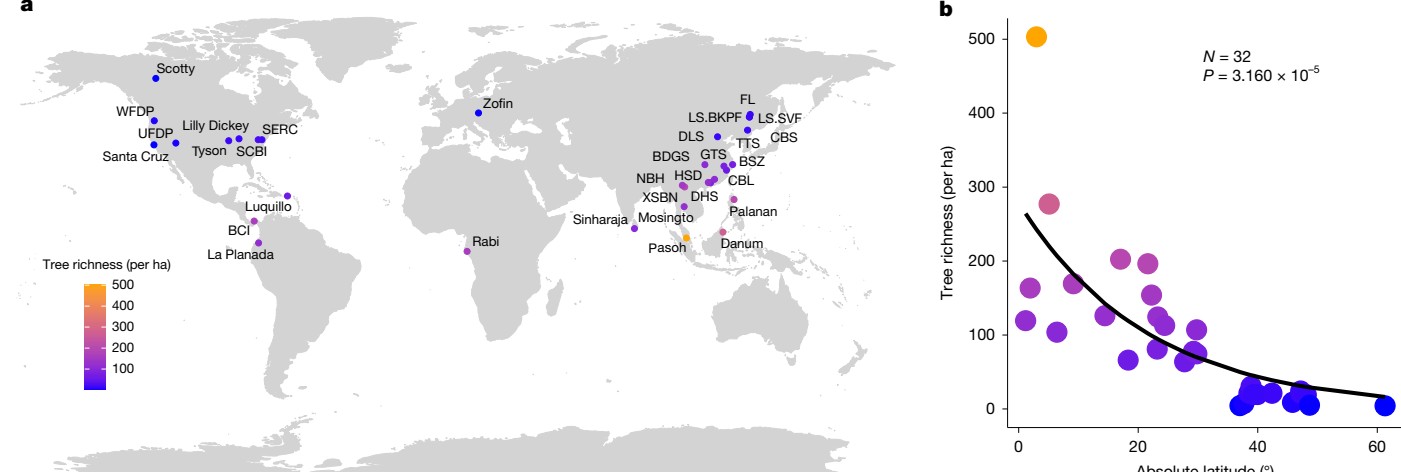

**Fig. 2 | Latitudinal tree diversity gradient across 32 forest plots worldwide. a**, Geographical distribution of the 32 forest plots. **b**, Latitudinal tree diversity gradient (number of tree species per hectare, distinguished by point colour). The latitudinal trend in tree diversity was fitted using an exponential

regression model; the solid line shows the fitted relationship. The significance of the regression coefficient was assessed using a two-sided $t$-test ($t = -4.892$, d.f. = 30, $P = 3.160 \times 10^{-5}$, $N = 32$ plots).

diversity gradient[3,4,12]. By contrast, our results show that facilitative pairwise interactions and HOIs are also common and strong (Fig. 4 and Extended Data Fig. 1), highlighting the necessity of considering their overall contributions (including both competitive and facilitative effects) to the latitudinal tree diversity gradient (Fig. 2b). Here we assessed the relative changes in growth rate caused by cumulative effects (both competitive and facilitative effects) of pairwise interactions and HOIs separately and examined how they changed with species abundance and latitude. The relative changes in growth rate caused by cumulative effects of pairwise interactions and HOIs both declined with species abundance, shifting from beneficial for rare species to detrimental for common species (Fig. 5). This pattern implies that both the cumulative effects of pairwise interactions and HOIs promote the growth rates of rare species but suppress those of common species, thereby potentially favouring species diversity at local scales. Moreover, the stabilizing effect of pairwise interactions

changed little across latitudes (the $P$ value for the interaction between abundance and latitude for the relative changes caused by pairwise interactions was 0.736; Table 1), whereas the stabilizing effect of HOIs became weaker (marginally significant change) towards higher latitudes (the $P$ value of the interaction between abundance and latitude for the relative change caused by HOIs was 0.093; Table 1). Taken together, these findings suggest that latitudinal changes in HOIs enhance the latitudinal tree diversity gradient, whereas latitudinal changes in pairwise interactions contribute little to this gradient.

Our study documents the prevalence of HOIs and their potential contribution to the latitudinal tree diversity gradient. For accurate interpretation of our results, we note some limitations of our analyses. First, the present study estimated pairwise interactions and HOIs separately for tree growth and survival (Extended Data Figs. 3–5 and Extended Data Table 1), thereby overlooking their covarying effects

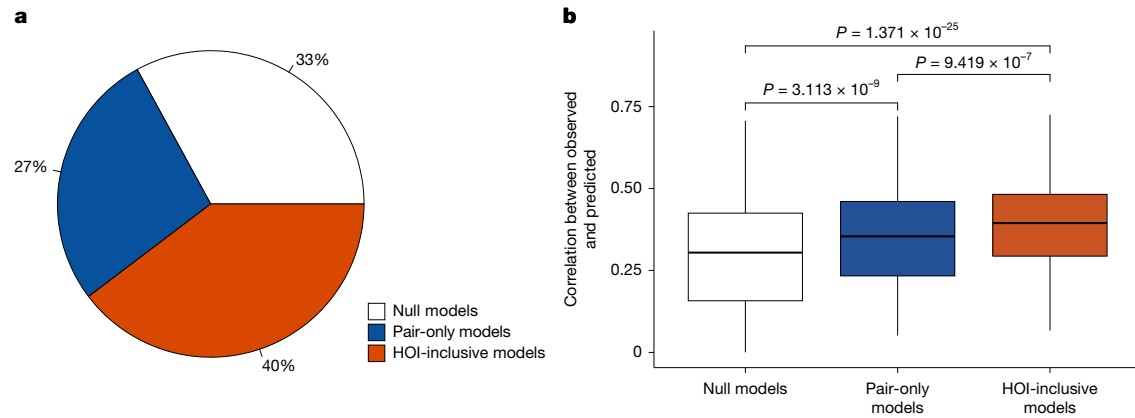

**Fig. 3 | Evidence of HOIs from growth models across 32 forest plots.**
**a**, Percentages of 1,543 tree species–plot combinations across the 32 plots for which each of the three classes of growth models were best supported (AIC at least two units lower than that of simpler models). The white, blue and orange slices indicate the percentages of species supporting the null models (models without including biotic interactions), pair-only models (models including only pairwise interactions) and HOI-inclusive models (models including both pairwise interactions and HOIs), respectively. **b**, Correlations between

observed and predicted growth rates by null, pair-only and HOI-inclusive models for the 609 species supported by the HOI-inclusive models. Box plots show the median (centre line) and 25th and 75th percentiles (box limits), with whiskers extending to the most extreme data points within 1.5× the interquartile range. Differences among models were assessed using pairwise two-sided $t$-tests ($N = 609$ species); exact $P$ values are shown in the figure. No adjustment was made for multiple comparisons.

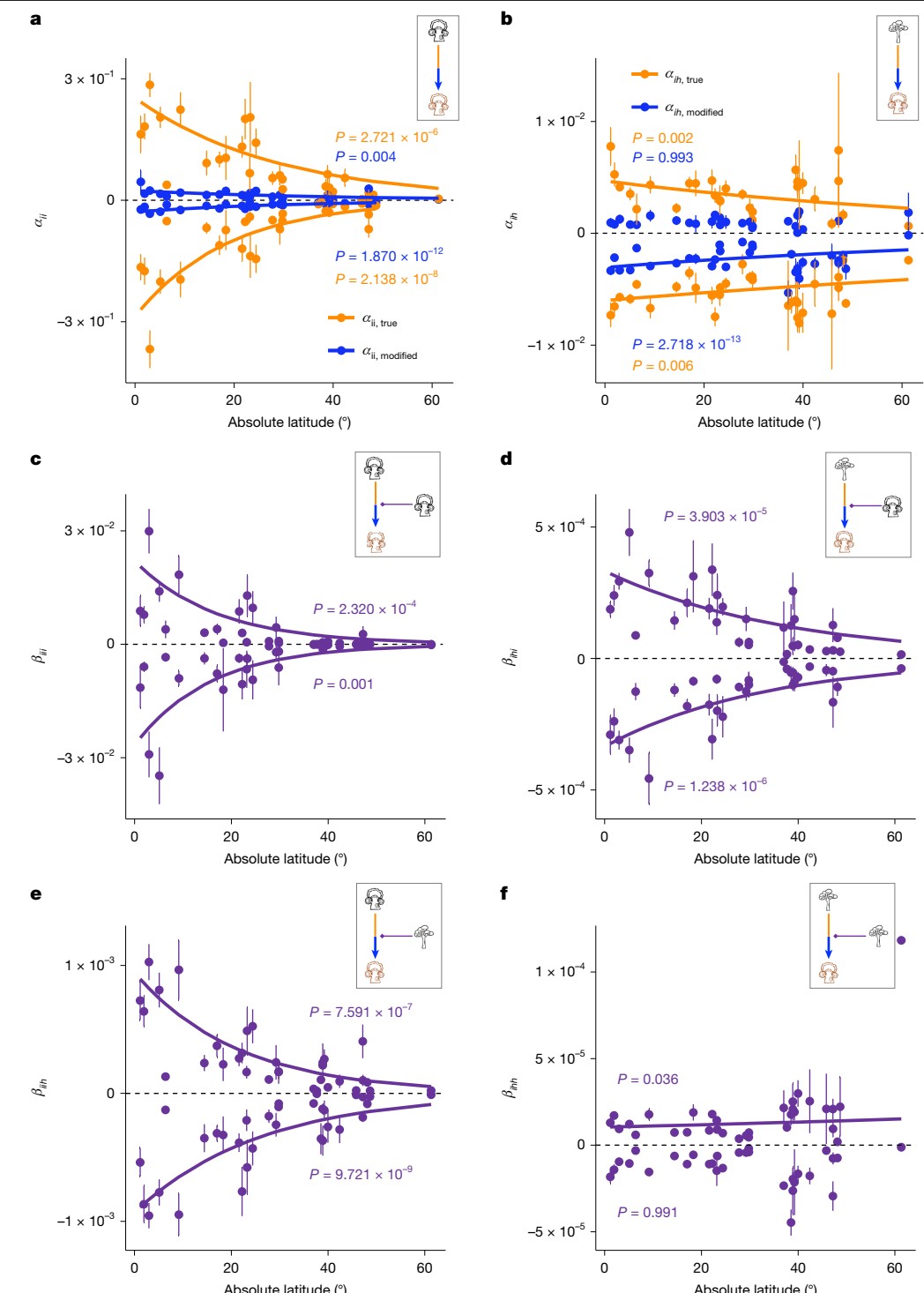

**Fig. 4 | Latitudinal changes in pairwise and HOIs for growth models.**
**a**,**b**, Latitudinal changes in intraspecific ($\alpha_{ii}$, **a**) and interspecific ($\alpha_{ih}$, **b**) pairwise interactions. The pairwise interactions estimated from pair-only models and HOI-inclusive models, denoted $\alpha_{modified}$ and $\alpha_{true}$, are distinguished by blue and orange points (lines). **c**,**e**, Latitudinal changes in HOI coefficients in which intraspecific pairwise interactions are modified by conspecific neighbours ($\beta_{iii}: \alpha_{ii} \leftarrow i$, **c**) and heterospecific neighbours ($\beta_{iih}: \alpha_{ii} \leftarrow h$, **e**), respectively. **d**,**f**, Latitudinal changes in HOI coefficients in which interspecific pairwise interactions are modified by conspecific neighbours ($\beta_{ihi}: \alpha_{ih} \leftarrow i$, **d**) and heterospecific neighbours ($\beta_{ihh}: \alpha_{ih} \leftarrow h$, **f**), respectively. The insets in each panel describe the six types of interaction (as in Fig. 1). Species-level pairwise interactions and HOIs were related to absolute latitude separately for competitive ($\alpha < 0, \beta < 0$) and facilitative ($\alpha > 0, \beta > 0$) interactions using exponential models. The significance of the regression coefficients was assessed using two-sided $t$-tests; exact $P$ values are shown in the figure. Regression lines are shown only when the interaction strength changed significantly with latitude ($P < 0.05$). For clarity, plot-level mean ± s.e.m. values are displayed rather than species-level estimates.

on population dynamics. Future studies should integrate multiple demographic processes (such as survival, growth and recruitment) to estimate overall species interactions influencing population

growth[34,35]. Second, pairwise interactions and HOIs estimated at the neighbourhood scale should be upscaled to the community scale before their contribution to species diversity can be rigorously

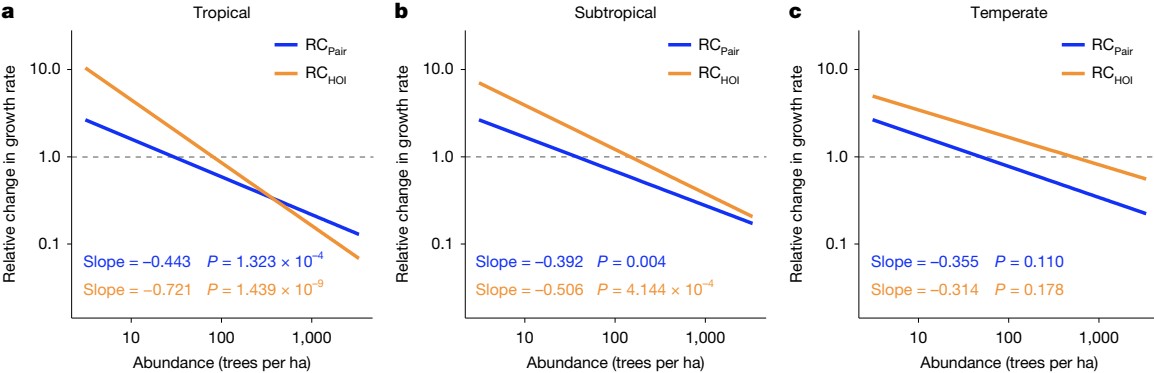

**Fig. 5 | Predicted relationships between relative change in growth rate and species abundance across three latitudinal geographic zones.** **a–c**, Relative changes in growth rate caused by cumulative effects of pairwise interactions (RC$_{Pair}$) and HOIs (RC$_{HOI}$) are distinguished by blue and orange lines, respectively. Predictions are shown for three geographic zones[3]: tropical (0–23.5°, **a**), subtropical (23.5–35.0°, **b**), and temperate (35.0–66.5°, **c**). Predictions were generated from the linear models summarized in Table 1, using the middle latitude of each zone (11.75° for the tropical zone, 29.25° for the subtropical zone and 45° for the temperate zone). Solid lines show model predictions, and shaded areas represent 95% confidence intervals.

assessed[36,37]. To stimulate further investigation, we conducted a preliminary upscaling analysis and explored the potential contribution of HOIs to species coexistence and the latitudinal diversity gradient based on structural stability and found that forest plots with higher species richness were less susceptible to loss of species (Supplementary Text 3). Third, we assumed that pairwise interactions would be modified by HOIs in a linear form, consistent with the findings of a broad range of previous theoretical[28,30,38] and empirical studies[33,39]. However, a recent study explored alternative functional forms and found that nonlinear formulation (using exponential and sigmoid functions) more accurately predicted population trends of annual plants than the linear form[40]; this warrants further investigation in forests.

How the latitudinal diversity gradient is maintained is a long-standing question in ecology and biogeography. We provide empirical evidence that HOIs are widespread in global forests. HOIs decline with latitude and weaken both the intraspecific (CNDD) and interspecific pairwise interactions, and thus enhance the latitudinal tree diversity gradient. Our findings help to resolve the debate on the role of pairwise CNDD in maintaining latitudinal diversity gradient and contribute to a shift from a paradigm of classical community ecology theory based solely on pairwise interactions towards a framework that the incorporates facilitations and HOIs.

## Table 1 | Summary of how relative changes in growth rate vary with species abundance and absolute latitude

| Response | Effect | Estimate | s.e. | t | P value |
|---|---|---|---|---|---|
| RC$_{Pair}$ | Intercept | 1.494 | 0.507 | 2.945 | 0.003 |
| | Abundance | −0.461 | 0.160 | −2.888 | 0.004 |
| | Latitude | −0.003 | 0.026 | −0.095 | 0.924 |
| | Abundance:latitude | 0.002 | 0.007 | 0.336 | 0.736 |
| RC$_{HOI}$ | Intercept | 3.586 | 0.531 | 6.752 | 2.07×10⁻¹¹ |
| | Abundance | −0.864 | 0.167 | −5.175 | 2.57×10⁻⁷ |
| | Latitude | −0.036 | 0.028 | −1.309 | 0.19 |
| | Abundance:latitude | 0.012 | 0.007 | 1.680 | 0.093 |

RC$_{Pair}$ and RC$_{HOI}$ represent relative changes in growth rate caused by cumulative effects of pairwise interactions and HOIs, respectively. The relative change in growth rate was fitted by linear models using log-transformed species abundance, absolute latitude and their interaction (N=1,543 species).

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

¹State Key Laboratory of Biocontrol, School of Ecology, Shenzhen Campus of Sun Yat-sen University, Shenzhen, China. ²Forest Global Earth Observatory, Smithsonian Tropical Research Institute, Panama City, Republic of Panama. ³School of BioSciences, University of Melbourne, Parkville, Victoria, Australia. ⁴Estación Biológica de Doñana, Consejo Superior de Investigaciones Científicas (EBD-CSIC), Seville, Spain. ⁵Center for Conservation and Sustainability, Smithsonian National Zoo and Conservation Biology Institute, Washington, DC, USA. ⁶Conservation Ecology Center, Smithsonian's National Zoo and Conservation Biology Institute, Front Royal, VA, USA. ⁷Department of Biology, Wilfrid Laurier University, Waterloo, Ontario, Canada. ⁸Department of Renewable Resources, University of Alberta, Edmonton, Alberta, Canada. ⁹IPHAMETRA, Herbier National du Gabon, Libreville, Gabon. ¹⁰National Biobank of Thailand, National Center for Genetic Engineering and Biotechnology, Pathum Thani, Thailand. ¹¹School of Biological Sciences, University of Aberdeen, Aberdeen, UK. ¹²Yunnan Key Laboratory of Forest Ecosystem Stability and Global Change, Xishuangbanna Tropical Botanical Garden, Chinese Academy of Sciences, Mengla, China. ¹³Department of Ecology and Evolutionary Biology, Tulane University, New Orleans, LA, USA. ¹⁴Forest Global Earth Observatory, Smithsonian Tropical Research Institute, Washington, DC, USA. ¹⁵State Key Laboratory of Vegetation and Environmental Change, Institute of Botany, Chinese Academy of Sciences, Beijing, China. ¹⁶Department of Science and Technology, Uva Wellassa University, Badulla, Sri Lanka. ¹⁷Gabon Biodiversity Program, Center for Conservation and Sustainability, Smithsonian National Zoo and Conservation Biology Institute, Gamba, Gabon. ¹⁸Department of Forest Biological Sciences, University of the Philippines Los Baños, Laguna, Philippines. ¹⁹Environmental Studies Department, University of California Santa Cruz, Santa Cruz, CA, USA. ²⁰School of Ecology and Environment, Northwestern Polytechnical University, Xi'an, China. ²¹Department of Forest Ecology, Landscape Research Institute, Brno, Czech Republic. ²²State Key Laboratory of Plant Diversity and Specialty Crops, Wuhan Botanical Garden, Chinese Academy of Sciences, Wuhan, China. ²³School of Ecology, Key Laboratory of Sustainable Forest Ecosystem Management-Ministry of Education, Northeast Asia Biodiversity Research Center, Northeast Forestry University, Harbin, China. ²⁴School of Forest, Fisheries, and Geomatics Sciences, University of Florida, Gainesville, FL, USA. ²⁵Natural Reserves, University of California Santa Cruz, Santa Cruz, CA, USA. ²⁶Department of Forest Management and Wilderness Institute, University of Montana, Missoula, MT, USA. ²⁷State Key Laboratory of Biocontrol, School of Life Science, Sun Yat-sen University, Guangzhou, China. ²⁸Guangdong Provincial Key Laboratory of Applied Botany, South China Botanical Garden, Chinese Academy of Sciences, Guangzhou, China. ²⁹Yunnan Academy of Forestry and Grassland, Kunming, China. ³⁰ECNU-Alberta Joint Lab for Biodiversity Study, Zhejiang Tiantong Forest Ecosystem National Observation and Research Station, School of Ecological and Environmental Science, East China Normal University, Shanghai, China. ³¹Department of Wildland Resources, Utah State University, Logan, UT, USA. ³²Zhejiang Qianjiangyuan Forest Biodiversity National Observation and Research Station, Key Laboratory of Vegetation and Environmental Change, Institute of Botany, Chinese Academy of Sciences, Beijing, China. ³³Smithsonian Environmental Research Center, Edgewater, MD, USA. ³⁴IRET/CENAREST, Libreville, Gabon. ³⁵Landscape Ecology Laboratory, Department of Landscape Architecture, University of Oregon, Eugene, OR, USA. ³⁶Department of Biology Washington University in St. Louis, St. Louis, MO, USA. ³⁷Forest Research Institute Malaysia, Kepong, Malaysia. ³⁸Estación Experimental de Zonas Áridas, Consejo Superior de Investigaciones Científicas, Almería, Spain. ³⁹IPHAMETRA, Arboretum de Sibang, Libreville, Gabon. ⁴⁰Department of Biology, Indiana University, Bloomington, IN, USA. ⁴¹Southeast Asia Rainforest Research Partnership (SEARRP), Kota Kinabalu, Malaysia. ⁴²Institute of Biology, College of Science, University of the Philippines Diliman, Quezon City, Philippines. ⁴³Zhejiang Tiantong Forest Ecosystem National Observation and Research Station, School of Ecological and Environmental Sciences, East China Normal University, Shanghai, China. ⁴⁴Guangdong Chebaling National Nature Reserve, Shaoguan, China. ⁴⁵Department of Forest Engineering, Resources, & Management, Oregon State University, Corvallis, OR, USA. ⁴⁶UK Centre for Ecology & Hydrology, Bush Estate, Penicuik, UK. ⁴⁷Department of Ecology, Evolution, and Environmental Biology, Columbia University, New York, NY, USA. ⁴⁸CAS Key Laboratory of Forest Ecology and Silviculture, Institute of Applied Ecology, Chinese Academy of Sciences, Shenyang, China. ⁴⁹State Key Laboratory for Vegetation Structure, Function and Construction (VegLab), College of Life Sciences, Zhejiang University, Hangzhou, China. ⁵⁰Department of Environmental Sciences, University of Puerto Rico, San Juan, PR, USA. ⁵¹These authors contributed equally: Yuanzhi Li, Junli Xiao. ⁵²e-mail: chuchjin@mail.sysu.edu.cn

## Methods

### Study sites and census data

Our study was based on multiple censuses of 32 large permanent forest dynamic plots, with an average plot size of 24.5 ha (range: 9–50 ha). Data were sourced from the Forest Global Earth Observatory (http://www.forestgeo.si.edu) and Chinese Forest Biodiversity Monitoring Network (http://www.cfbiodiv.org) (Fig. 2a and Supplementary Table 1). Plots span tropical to boreal terrestrial biomes with latitude ranging from 1.92° S to 61.30° N. All plots were established and censused several times following a standardized protocol[41]. In each census, all free-standing woody stems with a diameter at breast height (DBH) greater than 1 cm were tagged (unique ID), mapped (coordinates), identified (species identity), measured (DBH) and recorded (alive, dead or recruit). The census was repeated every 5 years to monitor forest dynamics (for instance, survival, growth and recruitment). Most plots were only censused twice. For a few plots with three or more censuses, we selected two consecutive censuses between 1998 and 2022 for analysis (Supplementary Table 1). Overall, we compiled data for more than 3 million trees of 5,000 species across the 32 study plots.

### Growth and survival models with HOIs

We estimated species interactions from demographic growth and survival models using the compiled dynamic forest census data[3,27]. For trees with more than one stem (that is, with multiple branches at height less than 1.3 m), we considered the survival and growth of the main stem. The growth of a focal tree $f$ of a species $i$ ($\text{Growth}_{i_f}$) was modelled as a function of its potential growth rate in the absence of neighbours ($G_i$), size ($\text{DBH}_{i_f}$), and neighbourhood pairwise ($\text{Pair}_{i_f}$) and higher-order effects ($\text{HOI}_{i_f}$)[27,42]:

$$\text{Growth}_{i_f} = G_i \times \text{DBH}_{i_f}^{\gamma} \times e^{\text{Pair}_{i_f}} \times e^{\text{HOI}_{i_f}}. \tag{1}$$

Similarly, the survival probability of a focal tree $f$ of a species $i$ ($\text{Survival}_{i_f}$) is modelled as[27]:

$$\text{Survival}_{i_f} = \frac{1}{1 + e^{\lambda_i + \gamma_1 \times \text{DBH}_{i_f}^{-1} + \gamma_2 \times \text{DBH}_{i_f} + \gamma_3 \times \text{DBH}_{i_f}^2 + \text{Pair}_{i_f} + \text{HOI}_{i_f}}}, \tag{2}$$

where $\lambda_i$ is the intrinsic survival probability in the absence of neighbours. The inverse of diameter ($\text{DBH}_{i_f}^{-1}$) is included to model rapid decline in mortality rate with increasing diameter, whereas the terms $\text{DBH}_{i_f}$ and $\text{DBH}_{i_f}^2$ model the U-shaped senescence effect[43].

The pairwise effects of all neighbours on the focal tree $i_f$ ($\text{Pair}_{i_f}$) can be decomposed into conspecific and heterospecific effects:

$$\text{Pair}_{i_f} = \alpha_{ii} \times n_{i,i_f} + \alpha_{ih} \times n_{h,i_f}, \tag{3}$$

where $\alpha_{ii}$ and $\alpha_{ih}$ are intraspecific and interspecific pairwise interaction coefficients, and $n_{i,i_f}$ and $n_{h,i_f}$ are conspecific and heterospecific pairwise neighbourhood crowding indices, respectively. Here we take a mean-field approximation[44] by replacing the species-specific interaction coefficients with a constant (average) heterospecific coefficient ($\alpha_{ih}$)[36,37]. The higher-order effects of all neighbours and all neighbours' neighbours on focal tree $i_f$ ($\text{HOI}_{i_f}$) can be decomposed into four components:

$$\text{HOI}_{i_f} = \beta_{iii} \times n_{ii,i_f} + \beta_{iih} \times n_{ih,i_f} + \beta_{ihi} \times n_{hi,i_f} + \beta_{ihh} \times n_{hh,i_f}, \tag{4}$$

where $\beta_{iii}$, $\beta_{iih}$, $\beta_{ihi}$ and $\beta_{ihh}$ are HOI coefficients associated respectively with intraspecific pairwise interactions modified by conspecific neighbours ($\beta_{iii} : \alpha_{ii} \leftarrow i$) or heterospecific neighbours ($\beta_{iih} : \alpha_{ii} \leftarrow h$), and interspecific pairwise interactions modified by conspecific neighbours ($\beta_{ihi} : \alpha_{ih} \leftarrow i$) or heterospecific neighbours ($\beta_{ihh} : \alpha_{ih} \leftarrow h$); and $n_{ii,i_f}$, $n_{ih,i_f}$, $n_{hi,i_f}$ and $n_{hh,i_f}$ are the corresponding higher-order neighbourhood crowding indices. The conspecific and heterospecific pairwise neighbourhood crowding indices $n_{i,i_f}$ and $n_{h,i_f}$ add up the crowding contributions of all conspecific and heterospecific neighbours, respectively, around a focal tree within a given radius. The contribution of a neighbour is directly proportional to its size and inversely proportional to the distance to the focal individual (equations S3 and S4 in Supplementary Text 1). The contribution of a neighbour to the corresponding higher-order crowding indices $n_{ii,i_f}$ and $n_{hi,i_f}$ is further multiplied by its conspecific crowding index (equations S7 and S9 in Supplementary Text 1), and that to $n_{ih,i_f}$ and $n_{hh,i_f}$ is further multiplied by its heterospecific crowding index (equations S8 and S10 in Supplementary Text 1). The calculation of pairwise and higher-order neighbourhood crowding indices is summarized in Supplementary Text 1 according to our previously developed method[27].

### Model fitting and evaluations

In each forest plot, we fitted demographic growth and survival models for each species with more than 100 trees (and also with at least 20 surviving and 20 dead trees for the survival model[3]) to ensure model performance and robustness. Overall, we fitted growth and survival models for 1,543 and 1,340 tree species–plot combinations, respectively. We log-transformed equation 1 to linearize the growth model and reduce model heteroscedasticity and residuals. We used logistic regression for tree survival (0 for death and 1 for alive). Overall, three classes of demographic models were constructed to determine the importance of tree size, pairwise interactions and HOIs: (1) null models without inclusion of biotic interactions, (2) pair-only models including pairwise interactions only, and (3) HOI-inclusive models including both pairwise interactions and HOIs. The pairwise interaction coefficients estimated from HOI-inclusive models were $\alpha_{\text{true}}$ (orange arrows in Fig. 1), because the effects of HOIs were isolated. By contrast, the pairwise coefficients estimated from pair-only models were $\alpha_{\text{modified}}$ (blue arrows in Fig. 1), because they implicitly incorporated effects of HOIs. Then, we compared the AIC values of the three models. The inclusion of HOIs was statistically supported when the AIC of the HOI-inclusive model was at least two units lower than those of the alternative models (Q1).

### Latitudinal changes in biotic interactions

To explore the latitudinal trend in biotic interactions (Q2), we fitted exponential relationships between the estimated coefficients of pairwise interactions ($\alpha_{ii}$ and $\alpha_{ih}$) and HOIs ($\beta_{iii}$, $\beta_{iih}$, $\beta_{ihi}$ and $\beta_{ihh}$) and absolute latitude for competitive ($\alpha < 0$, $\beta < 0$) and facilitative ($\alpha > 0$, $\beta > 0$) interactions separately, given that they were nearly equally frequent (Extended Data Fig. 1). We further evaluated how HOIs ($\beta_{ijk}$) modified the true pairwise interactions ($\alpha_{ij,\text{true}}$) by examining their correlations.

### Contribution of biotic interactions to latitudinal diversity gradient

To further assess how the latitudinal change in pairwise interactions and HOIs might contribute to the latitudinal tree diversity gradient (Q3), we first calculated the relative change in growth rate and survival probability caused by the neighbourhood cumulative effects of true pairwise interactions ($e^{\text{Pair}_{i_f}}$) and HOIs ($e^{\text{HOI}_{i_f}}$) separately for each focal tree (equations 1–4)[27,33] and then averaged the relative change across trees for each species ($\text{RC}_{\text{Pair}} = \sum_{f=1}^{N_i} e^{\text{Pair}_{i_f}}/N_i$, $\text{RC}_{\text{HOI}} = \sum_{f=1}^{N_i} e^{\text{HOI}_{i_f}}/N_i$). Finally, we explored how the relative changes ($\text{RC}_{\text{Pair}}$ and $\text{RC}_{\text{HOI}}$) varied with species abundance per hectare and absolute latitude using linear regression. The relative changes and species abundance were log-transformed to improve normality.

Results from the survival models (Extended Data Figs. 3–5 and Extended Data Table 1) were qualitatively consistent with those from the growth models. Therefore, we only reported the growth model results in the main text. Moreover, as demonstrated in Supplementary Text 2, the growth model results were robust to variations in parameter settings (for instance, for different neighbourhood

radii), inclusion of spatial autocorrelation (with or without quadrats as random effects), uncertainty in estimation of interaction coefficients, and the distinction between small (DBH < 10 cm) and large (DBH ≥ 10 cm) trees. All analyses were conducted in R v.4.4.1 (ref. 45). The map in Fig. 2 was generated using R packages ggplot2 (v.4.0.0) and ggrepel (v.0.9.6).

## Reporting summary

Further information on research design is available in the Nature Portfolio Reporting Summary linked to this article.

## Data availability

The raw census data that support this study are available upon request and with permission of the principal investigators of the Forest Global Earth Observatory and Chinese Forest Biodiversity Monitoring Network networks (names and contact information of the principal investigators are provided in Supplementary Table 1). For some plots, the data are publicly available at https://forestgeo.si.edu/explore-data. The processed datasets supporting the findings of this study are publicly available at Figshare[46] (https://doi.org/10.6084/m9.figshare.28426862).

## Code availability

The custom code used in this study is available at Figshare[46] (https://doi.org/10.6084/m9.figshare.28426862).

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

**Acknowledgements** We thank J. Levine for insightful comments on this manuscript, and T. Wiegand for help with upscaling the interaction coefficients from neighbourhood to population scale. This research was funded by grants from the National Natural Science Foundation of China (32330064, 32271595, 32401280, 32525006, 31925027), the National Key Research and Development Programme of China (2022YFF0802300), the Fundamental and Interdisciplinary Disciplines Breakthrough Plan of the Ministry of Education of China (JYB2025XDXM902), and the Guangdong Provincial Field Observation and Research Station for Biodiversity and Biotic Interactions in Chebaling Lingnan Mountain Forests (2025B1212050003). Funding and references related to each forest plot are listed in Supplementary Table 1.

**Author contributions** Y. Li, C.C. and J.X. conceived the study in conversation with M.M.M. Y. Li and C.C. assembled the forest census data. Y.J. and Y. Li cleaned and formatted the data. J.X. and Y. Li analysed the data and generated figures and tables. Y. Li wrote the manuscript with input from C.C., J.X., S.J.W., M.M.M. and O.G. F.H. made substantial contributions during revisions. The authors (including A.A., K.J.A.-T., J.B., J.D.B., P.B., N.A.B., W.B., D.F.R.P.B., M.C., K.C., S.J.D., Q.D., S.E., A.F., E.S.F., G.S.G., Z.H., J.H., M.J., G.J., D.J.J., A.S.J., K.K., A.J.L., B.L., J.L., L.L., F.L., Y. Liu, Z.L., J.A.L., K.M., S.M.M., W.M., H.R.M., X.M., J.A.M., M.N., A.N., M.J.O., N.L.E.O., G.P., R.P.P., X.Q., H.R., G.R., L.J.V.R., P.Š., G.S., Z.S., J.S., M.E.S., J.T., M.U., Xihua Wang, Xugao Wang, Y.W., T.L.Y., W.Y., M.Y., M.Z., Y.Z., J.Z., F.H. and C.C.) contributed forest census data and revised the manuscript.

**Competing interests** The authors declare no competing interests.

**Additional information**
**Correspondence and requests for materials** should be addressed to Chengjin Chu.

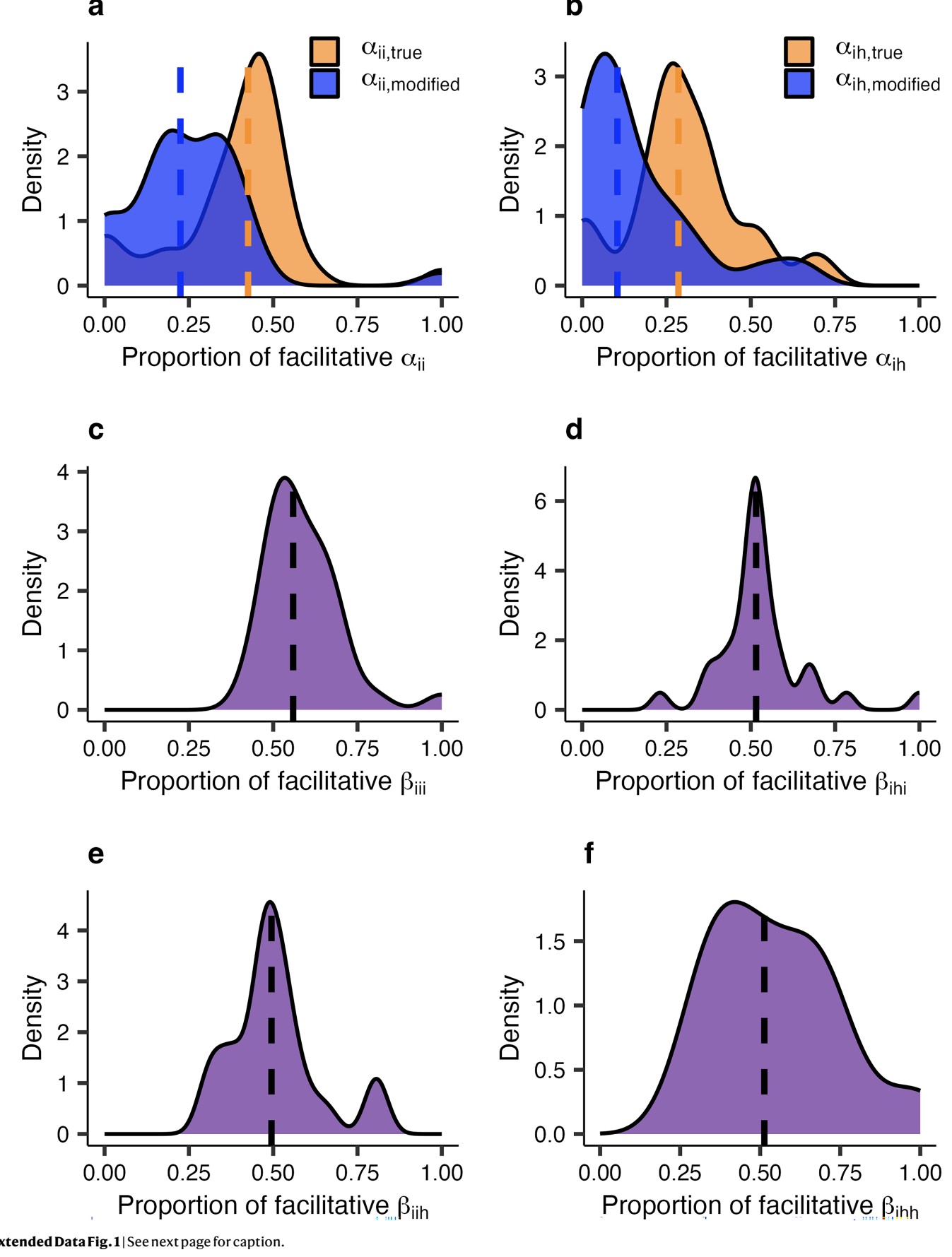

**Extended Data Fig. 1** | See next page for caption.

**Extended Data Fig. 1 | Proportion of facilitative pairwise and higher-order interactions estimated from growth models.** Panels (a) and (b) show the distribution of proportion of facilitative intraspecific ($\alpha_{ii}$) and interspecific ($\alpha_{ih}$) pairwise interactions, respectively. The pairwise interactions estimated from PAIR-only models and HOI-inclusive models, denoted as $\alpha_{\text{modified}}$ and $\alpha_{\text{true}}$, are distinguished by blue and orange color. Panels (c) and (e) show the distribution of proportion of facilitative higher-order interaction coefficients $\beta_{iii}$ and $\beta_{iih}$, which represent the modifications of intraspecific pairwise interactions by conspecific neighbors ($\alpha_{ii} \leftarrow i$) and by heterospecific neighbors ($\alpha_{ii} \leftarrow h$), respectively. Panels (d) and (f) show the distribution of proportion of facilitative higher-order interaction coefficients $\beta_{ihi}$ and $\beta_{ihh}$, corresponding to the modifications of interspecific pairwise interactions by conspecific neighbors ($\alpha_{ih} \leftarrow i$) and heterospecific neighbors ($\alpha_{ih} \leftarrow h$), respectively.

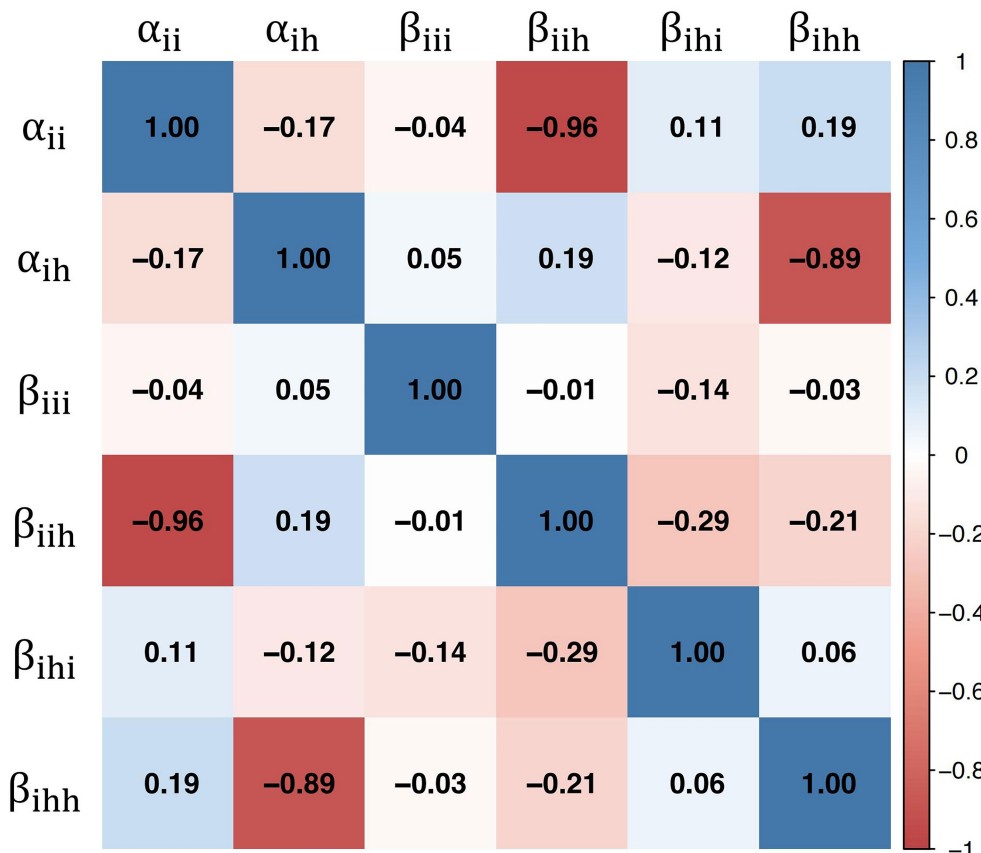

**Extended Data Fig. 2 | Pearson correlation coefficients between different types of interactions estimated from growth models.** Here, $\alpha_{ii}$ and $\alpha_{ih}$ denote intraspecific and interspecific pairwise interactions estimated from HOI-inclusive models. $\beta_{iii}$, $\beta_{iih}$, $\beta_{ihi}$ and $\beta_{ihh}$ are higher-order interaction coefficients referring to intraspecific pairwise interactions modified by conspecific neighbors ($\alpha_{ii} \leftarrow i$), intraspecific pairwise interactions modified by heterospecific neighbors ($\alpha_{ii} \leftarrow h$), interspecific pairwise interactions modified by conspecific neighbors ($\alpha_{ih} \leftarrow i$), and interspecific pairwise interactions modified by heterospecific neighbors ($\alpha_{ih} \leftarrow h$), respectively.

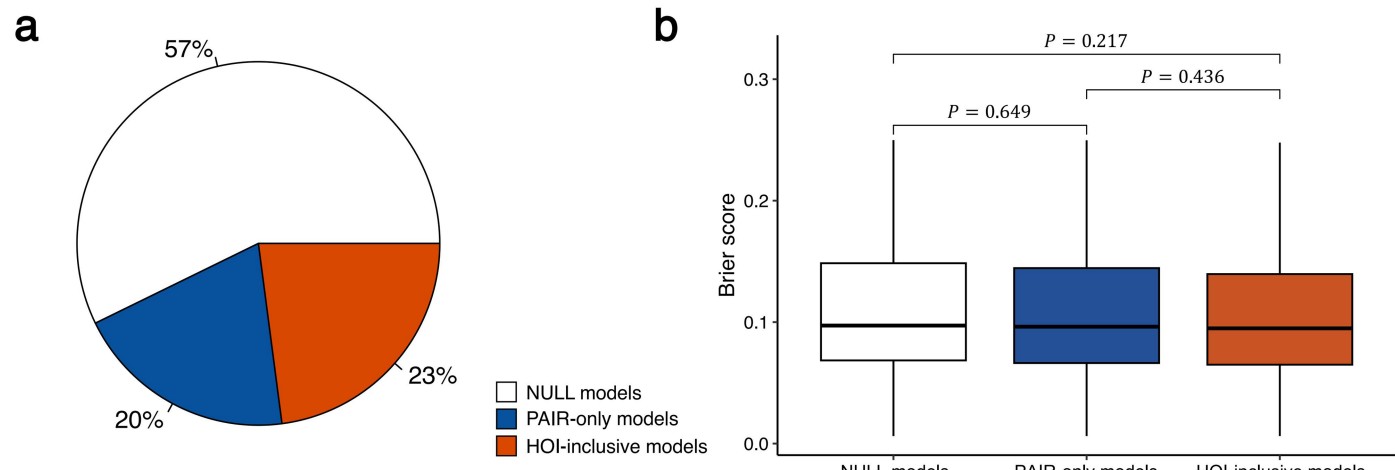

**Extended Data Fig. 3 | Evidence of higher-order interactions from survival models across 32 forest plots.** Panel (a) shows the percentage of 1,340 tree species–plot combinations across the 32 plots for which each of the three classes of survival models was best supported (AIC at least 2 units lower than that of the two alternative models). The white, blue and orange regions indicate the percentage of species supporting the NULL models (models without including biotic interactions), PAIR-only models (models includes only pairwise interactions) and the HOI-inclusive models (models include both pairwise interactions and HOIs), respectively. Panel (b) is box-plot of the Brier score between observed survival outcome and predicted survival probabilities by NULL, PAIR-only and HOI-inclusive models for the 306 species supported by the HOI-inclusive models. Brier score is calculated as $\frac{1}{n}\sum_{i=1}^{n}(y_i - p_i)^2$, where $y_i$ is the observed survival outcome (0 for death and 1 for alive), $p_i$ is the predicted survival probability for observation $i$, and $n$ is the sample size for each species. A lower Brier score indicates more accurate predictions. Box plots show the median (center line), the 25th and 75th percentiles (box limits), and whiskers extending to the most extreme data points within 1.5 × the interquartile range. Differences among models are assessed using pairwise two-sided $t$-test ($N$ = 306 species); exact $P$ values are shown in the figure. No adjustment is made for multiple comparisons.

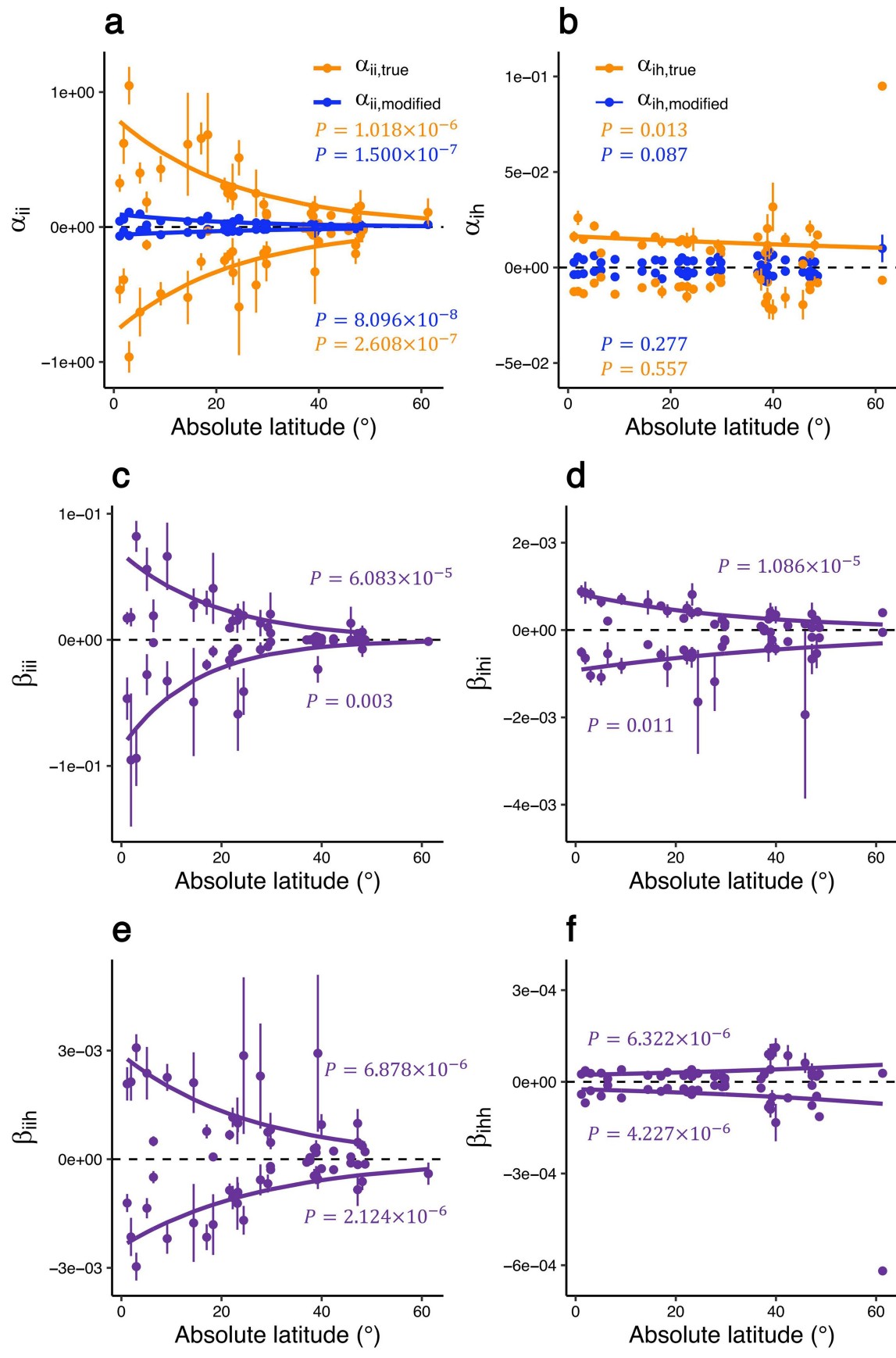

**Extended Data Fig. 4** | See next page for caption.

**Extended Data Fig. 4 | Latitudinal changes in pairwise and higher-order interactions for survival models.** Panels (a) and (b) display the latitudinal changes in intraspecific ($\alpha_{ii}$) and interspecific ($\alpha_{ih}$) pairwise interactions, respectively. The pairwise interactions estimated from PAIR-only models and HOI-inclusive models, denoted as $\alpha_{\mathrm{modified}}$ and $\alpha_{\mathrm{true}}$, are distinguished by blue and orange points (lines). Panels (c) and (e) show the latitudinal changes in higher-order interaction coefficients in which intraspecific pairwise interactions are modified by conspecific neighbors ($\beta_{iii}$) and by heterospecific neighbors ($\beta_{iih}$), respectively. Panels (d) and (f) show the latitudinal changes in higher-order interaction coefficients in which interspecific pairwise interactions are modified by conspecific neighbors ($\beta_{ihi}$) and by heterospecific neighbors ($\beta_{ihh}$), respectively. The insets in each panel describe the six types of interaction (see Fig. 1). Species-level pairwise and higher-order interactions were related to absolute latitude separately for competitive ($\alpha < 0, \beta < 0$) and facilitative interactions ($\alpha > 0, \beta > 0$) using exponential models. Significance of the regression coefficients is assessed using two-sided $t$-test; exact $P$ values are shown in the figure. Regression lines are shown only when the interaction strength changed significantly with latitude ($P < 0.05$). For clarity, we displayed plot-level mean values +/− SEM rather than species-level estimates.

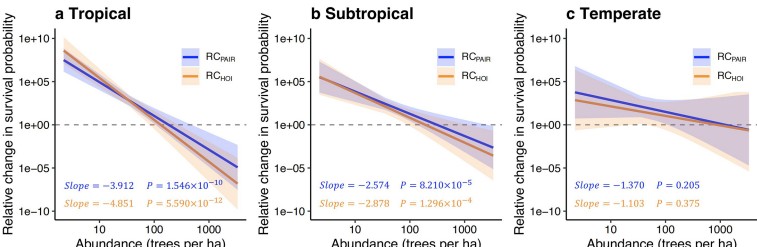

**Extended Data Fig. 5 | Predicted relationships between relative change in survival probability and species abundance across three latitudinal geographic zones.** $RC_{PAIR}$ and $RC_{HOI}$ that represent relative changes in the odds of survival ($\frac{p}{1-p}$, $p$ is survival probability) caused by cumulative effects of pairwise and higher-order interactions, are distinguished by blue and orange lines. Predictions are shown for three geographic zones: tropical zone (0°–23.5°, a), subtropical zone (23.5°–35°, b), and temperate zone (35°–66.5°, c). Predictions are generated from the linear models summarized in Extended Data Table 1, using the middle latitude of each zone (i.e., 11.75° for tropical zone, 29.25° for subtropical zone and 45° for temperate zone). Solid lines show model predictions and shaded areas represent 95% confidence intervals.

**Extended Data Table 1 | Summary of relative changes in survival probability varying with species abundance and absolute latitude**

| Response | Effect | Estimate | Standard error | *t* value | *P* value |
|---|---|---|---|---|---|
| RC$_{PAIR}$ | Intercept | 24.136 | 2.977 | 8.106 | $1.17 \times 10^{-15}$ |
| | Abundance | -4.810 | 0.884 | -5.438 | $6.39 \times 10^{-8}$ |
| | Latitude | -0.320 | 0.138 | -2.322 | 0.020 |
| | Abundance:Latitude | 0.076 | 0.036 | 2.106 | 0.035 |
| RC$_{HOI}$ | Intercept | 29.378 | 3.427 | 8.574 | $2.73 \times 10^{-17}$ |
| | Abundance | -6.176 | 1.018 | -6.068 | $1.68 \times 10^{-9}$ |
| | Latitude | -0.487 | 0.158 | -3.075 | 0.002 |
| | Abundance:Latitude | 0.113 | 0.042 | 2.699 | 0.007 |

RC$_{PAIR}$ and RC$_{HOI}$ represent relative changes in the odds of survival ($\frac{p}{1-p}$, $p$ is survival probability) caused by cumulative effects of pairwise and higher-order interactions, respectively. The relative change in growth rate is fitted by linear models using log-transformed species abundance, absolute latitude and their interaction as predictors ($N$=1,340 species).

# Reporting Summary

## Statistics

For all statistical analyses, confirm that the following items are present in the figure legend, table legend, main text, or Methods section.

| n/a | Confirmed | |
|---|---|---|
| ☐ | ☒ | The exact sample size (*n*) for each experimental group/condition, given as a discrete number and unit of measurement |
| ☐ | ☒ | A statement on whether measurements were taken from distinct samples or whether the same sample was measured repeatedly |
| ☐ | ☒ | The statistical test(s) used AND whether they are one- or two-sided *Only common tests should be described solely by name; describe more complex techniques in the Methods section.* |
| ☐ | ☒ | A description of all covariates tested |
| ☐ | ☒ | A description of any assumptions or corrections, such as tests of normality and adjustment for multiple comparisons |
| ☐ | ☒ | A full description of the statistical parameters including central tendency (e.g. means) or other basic estimates (e.g. regression coefficient) AND variation (e.g. standard deviation) or associated estimates of uncertainty (e.g. confidence intervals) |
| ☐ | ☒ | For null hypothesis testing, the test statistic (e.g. *F*, *t*, *r*) with confidence intervals, effect sizes, degrees of freedom and *P* value noted *Give P values as exact values whenever suitable.* |
| ☒ | ☐ | For Bayesian analysis, information on the choice of priors and Markov chain Monte Carlo settings |
| ☒ | ☐ | For hierarchical and complex designs, identification of the appropriate level for tests and full reporting of outcomes |
| ☐ | ☒ | Estimates of effect sizes (e.g. Cohen's *d*, Pearson's *r*), indicating how they were calculated |

*Our web collection on statistics for biologists contains articles on many of the points above.*

## Software and code

Policy information about availability of computer code

| Data collection | No software was used for data collection. |
|---|---|
| Data analysis | Data analyses were conducted in R version 4.4.1 using several packages, including DHARMa (v0.4.7), minpack.lm (v1.2-4) and ggplot2(v4.0.0). Additional packages used in the analyses are documented in the code. The custom code used in this study is available at Figshare: https://doi.org/10.6084/m9.figshare.28426862. |

For manuscripts utilizing custom algorithms or software that are central to the research but not yet described in published literature, software must be made available to editors and reviewers. We strongly encourage code deposition in a community repository (e.g. GitHub). See the Nature Portfolio guidelines for submitting code & software for further information.

## Data

Policy information about availability of data

All manuscripts must include a data availability statement. This statement should provide the following information, where applicable:

- Accession codes, unique identifiers, or web links for publicly available datasets
- A description of any restrictions on data availability
- For clinical datasets or third party data, please ensure that the statement adheres to our policy

The raw census data that support this study are available upon request and with permission of the principal investigators of the ForestGEO and CForBio networks

# Research involving human participants, their data, or biological material

Policy information about studies with [human participants or human data](). See also policy information about [sex, gender (identity/presentation), and sexual orientation]() and [race, ethnicity and racism]().

| Reporting on sex and gender | Not applicable. |
|---|---|
| Reporting on race, ethnicity, or other socially relevant groupings | Not applicable. |
| Population characteristics | Not applicable. |
| Recruitment | Not applicable. |
| Ethics oversight | Not applicable. |

Note that full information on the approval of the study protocol must also be provided in the manuscript.

# Field-specific reporting

Please select the one below that is the best fit for your research. If you are not sure, read the appropriate sections before making your selection.

☐ Life sciences          ☐ Behavioural & social sciences          ☒ Ecological, evolutionary & environmental sciences

For a reference copy of the document with all sections, see [nature.com/documents/nr-reporting-summary-flat.pdf]()

# Ecological, evolutionary & environmental sciences study design

All studies must disclose on these points even when the disclosure is negative.

| Study description | In this study, we assembled census data from 32 large permanent forest plots spanning tropical to boreal forests to address three key questions: (Q1) Are HOIs prevalent among trees across forest plots? (Q2) How do pairwise interactions and HOIs vary with latitude? (Q3) How do latitudinal changes in HOIs contribute to the latitudinal tree diversity gradient? To answer these questions, we estimated both pairwise interactions and HOIs from demographic growth (for 1,543 tree species–plot combinations) and survival models (for 1,340 tree species–plot combinations), respectively. With these data, we built three types of growth and survival models: (i) NULL models with no biotic interactions, (ii) PAIR-only models including only pairwise interactions, and (iii) HOI-inclusive models including both pairwise interactions and HOIs. We then compared AIC of the three types of models (Q1), tested whether the estimated pairwise interactions and HOIs declined with latitudes (Q2), and evaluated how the cumulative effects of pairwise interactions and HOIs on growth and survival changed with species abundance and latitude (Q3). |
|---|---|
| Research sample | The data used in this study were collected at 32 large permanent forest dynamic plots worldwide from the Forest Global Earth Observatory (ForestGEO, http://www. forestgeo.si.edu) and Chinese Forest Biodiversity Monitoring Network (CForBio, http://www.cfbiodiv.org). |
| Sampling strategy | Most plots had only been censused twice. For a few plots with three or more censuses, we selected two consecutive censuses between 1998 and 2022 for analysis. Overall, we compiled data for over three million trees of 5,000 species across 32 plots. |
| Data collection | All plots were established and censused multiple times following a standardized protocol. In each census, all free-standing woody stems with a diameter at breast height (DBH) larger than 1 cm were tagged (unique ID), mapped (coordinates), identified (species identity), measured (DBH), and recorded (alive, dead or recruit). The census was repeated every 5 years to monitor forest dynamics (e.g., survival, growth, and recruitment). |
| Timing and spatial scale | The 32 forest plots span tropical to boreal terrestrial biomes with latitude ranging from 1.92° S to 61.30° N. These plots vary in size between 9 and 50 ha (24.5 ha on average). At all plots included in this study, two ore more censuses have been carried out with remeasurement intervals of approximately five years. Most plots had only been censused twice. For a few plots with three or more censuses, we selected two consecutive censuses between 1998 and 2022 for analysis. |
| Data exclusions | Observations of trees were excluded when information on coordinates, species, or status was missing. We fit demographic growth and survival models for each species with more than 100 trees (and additionally with at least 20 alive and dead observations for the survival model) to ensure model performance and robustness. |
| Reproducibility | This study is based on observational data, and no experiments are conducted. |
| Randomization | This study is based on observational data. Therefore, randomization into groups does not apply. |

| | Blinding | Blinding is not relevant for this study because the data were not specifically collected to assess direct and higher-order interactions. |

Did the study involve field work? ☐ Yes ☒ No

# Reporting for specific materials, systems and methods

We require information from authors about some types of materials, experimental systems and methods used in many studies. Here, indicate whether each material, system or method listed is relevant to your study. If you are not sure if a list item applies to your research, read the appropriate section before selecting a response.

## Materials & experimental systems

| n/a | Involved in the study |
|---|---|
| ☒ | ☐ Antibodies |
| ☒ | ☐ Eukaryotic cell lines |
| ☒ | ☐ Palaeontology and archaeology |
| ☒ | ☐ Animals and other organisms |
| ☒ | ☐ Clinical data |
| ☒ | ☐ Dual use research of concern |
| ☒ | ☐ Plants |

## Methods

| n/a | Involved in the study |
|---|---|
| ☒ | ☐ ChIP-seq |
| ☒ | ☐ Flow cytometry |
| ☒ | ☐ MRI-based neuroimaging |

## Plants

| Seed stocks | Not applicable. |

| Novel plant genotypes | Not applicable. |

| Authentication | Not applicable. |

