## [Peer Review File · Nature]

Higher-order interactions enhance the latitudinal tree diversity gradient

Corresponding Author: Professor Chengjin Chu

Version 0:

Reviewer comments:

Referee #1

(Remarks to the Author)

This study uses a large data set of fully mapped forest inventory plots around the globe to study the effects of pairwise and higher-order interactions on growth and survival of individual trees. The objectives are to assess if higher-order interactions play globally an important role in governing growth and survival in forests, if pairwise and higher-order interactions change systematically along latitude, and if latitudinal gradients in higher-order interactions would be related to the latitudinal gradient in tree diversity.

The study is a long-needed contribution that may change the way species interactions in plant communities are conceptualized. The authors compiled an impressive data set to address their objectives. However, while I see a large potential in the study, the current analyses and descriptions go often not deep enough and need some revision. I have three major comments and a number of smaller comments and suggestions to improve the work.

1) To improve the biological interpretation of the analyses I propose that the authors conduct separate analyses for juveniles and for reproductive trees or for large trees (dbh \geq 10cm) and small trees (dbh $<$ 10cm). Because the vast majority of all trees in your analysis will be very small trees your current results are heavily driven by small trees, but for demographic assessment you need to deal with larger reproductive trees. In any case, it would be interesting if the HOI (and pairwise) effects change with tree size.

2) The section on structural stability is the weakest and most cryptic part of your analysis. While I understand that it is tempting to investigate the consequences of the interesting patterns found, it leaves the impression that the concepts have not yet been sufficiently matured for the analyses you want to carry out. I therefore propose to remove it and replace it by a deeper biological analysis of pairwise and HOT interactions, e.g., by investigating how the importance of HOI and pairwise effects change with tree size. Regarding the structural stability analysis, it is incorrect to use the interaction coefficients of a neighborhood-scale analysis in a community-scale demographic model, you need to conduct an additional upscaling step to translate the interactions at the scale of individual trees to Lotka-Volterra interactions coefficients.

3) I like to see a correlation analysis of the quantities PAIR_{ij} and HOI_{ijh} and the quantities PAIR_{ih} and HOI_{ihh}. If they are strongly correlated (as I suspect), one of each must be removed from the model. Conceptually, I doubt that it is correct to include both pairwise and HOI interactions in the survival and growth models. My argument is that the HOIs basically refine the pairwise interactions by allowing a neighbor to compete more or less than the average (assumed by pairwise interactions), depending on the crowding it suffers itself.

Specific comments

line 131-133

To estimate $\alpha_{ij,true}$, did you select only pairs of individuals without other neighbors, as suggested by the text: "the effect of a neighbor tree j on a target tree i in the absence of other trees"? Or is $\alpha_{(ij,true)}$ the coefficient determined in the analysis that includes both pairwise and HOI effects? I notice that you briefly define $\alpha_{ij,true}$ and $\alpha_{ij,modified}$ in lines 168-170, but this terminology must be made much clearer through the text.

Figure 1

I think panels b and c are slightly misleading since the presence of neighbors k of neighbor j modifies the interaction strength of neighbor j on the focal individual i. So, the arrows should go from k to j.

line 133-136

This description does not really capture what you analyze here in forest systems where one focal tree has usually many neighbors. Alternatively, you could explain that the pairwise approach assumes that the performance (growth, survival) of a focal tree depends on the neighborhood crowding surrounding the focal individual i (that sums up the contribution of each con- and heterospecific neighbor i_p and i_h within a given distance R of the focal individual). However, the HOI approach assumes that the contribution of the neighbors i_p and i_h may be somewhat larger or smaller than assumed by the pairwise approach if one considers the crowding indices of the neighbors i_p and i_h . In case of competitive interactions, for example, a neighbor i_p may compete more weakly if it suffers large than average neighborhood crowding, and it may compete more strongly if it enjoys a smaller than average neighborhood crowding.

line 138

Here you should cite the original study and provide the Latin names of the species. Note that the term "plantain" refers also to cooking banana, which is probably not what you mean.

lines 146-148

You should add here that one should first investigate the effect of HOIs on plant performance (e.g., growth and survival) before targeting coexistence.

Line 148

Perhaps better: "the recently developed structural approach for understanding multispecies coexistence"

lines 146-158

This is the weakest part of the manuscript, see my comments to the methods below.

lines 160-163

The aims are somewhat unspecific with respect to the underlying biology. I guess that you conduct your estimates for all focal individuals in the forest plots, without considering size or reproductive status. Given that the vast majority of all trees will be very small trees (say $dbh < 2.5$ cm) your results will be heavily driven by small trees, but a demographic Lotka-Volterra model (used to derive structural stability) deals with reproductive trees. So, I like to see separate analyses for juveniles and for reproductive trees or at least for large trees ($dbh \geq 10$ cm) and small trees ($dbh < 10$ cm).

lines 163-172

This is a nice approach, however, did you consider models with $\Delta_{AIC} < 2$ to be equally supported by the data? You mention only that you select the model with the lowest AIC. Please clarify. If the AIC of pairwise and HOI differ by less than two, you should select the simpler model (the pairwise one).

lines 172-174

There is a scaling issue when you apply your interaction coefficients to structural stability. You estimate the pairwise interaction coefficients and the HOIs at the plant neighborhood scale, but to get the appropriate interaction coefficients at the community scale to be fed into Lotka-Volterra models you need to upscale (see reference 42).

lines 176-178

If the AIC of pairwise and HOI models differ by less than two, you should select the simpler model (the pairwise one).

Here the reader needs to get an idea of how much better the HOI-inclusive models are in predicting growth and survival compared to the next best model. For growth you could show observed vs. predicted plots.

The analyses should be repeated for juveniles and reproductive (and/or for small and large) trees.

lines 180-181

This interesting result is driven by small trees, what happens for larger trees?

line 184-191

I am not sure if you can interpret positive coefficients generally as facilitation. Especially juvenile trees are often clustered together in smaller multispecies clumps e.g., caused by canopy gaps or because they are dispersed together by animals. If these sites are favorable microsites (where saplings grow and survive better than in the surrounding), this may also result in positive coefficients.

Fig. 3.

The decline in the average positive and negative coefficients with latitude is an interesting pattern!

Line 194-202

The correlations between α_{ij} and β_{ijh} and α_{ih} and β_{ihh} are quite high. I am not entirely convinced by your interpretation. Did you check if the different variables in your model are correlated? With equations 5 and 6 you have six

interaction terms in your equations 1 and 2, I guess that the terms PAIR_{ij} and HOI_{ijh} as well as the terms PAIR_{ih} and HOI_{ihh} would be strongly correlated. If pairs of PAIR and HOI terms are strongly correlated, you need to remove one of them from the model.

Lines 411

To help the reader, I would simplify the equations by setting $u = 2$ and $v = 1$ and $m = 0$ (similar to of line 456), but mention at an appropriate place that you tested additional scaling parameters u and v and saturation parameters m . I would use $u = 2$, because this means that species interactions would be mediated by basal area (which scales with biomass). Additionally, to be consistent with the established terminology I would suggest that you call the quantities PAIR_{if} neighborhood crowding indices (or neighborhood competition indices) and cite one of the studies that developed this approach (e.g., Canham et al., 2004: doi: 10.1139/X03-232; Uriarte et al., 2004: doi: 10.1111/j.0022-0477.2004.00867.x). It may also help to express equation 6 in terms of the neighborhood crowding indices of the con- and heterospecific second-order neighbors i_p and h_p of the focal tree i_f .

Lines 427-432

The grouping into con- and heterospecifics is usually called “mean field approximation” (e.g., O’Dwyer & Chisholm 2014: Ecol. Lett. 17, 961–969) and rests on the assumption of diffuse competition, where the focal species interacts with an average taken over other species in the community. It rests on the diversity and variability of the local surroundings of each individual.

Lines 435-442

By grouping species into con- and heterospecifics the initial idea of species-specific higher-order interactions is somewhat lost. You should mention the con- vs heterospecific grouping earlier on in the main text to avoid wrong expectations of readers.

Lines 458-461

Given that the vast majority of all trees you analyzed will be very small trees (say $dbh < 2.5$ cm) your results will be heavily driven by small trees if you use all individuals together. However, small saplings and adult trees may behave quite differently. To make your analysis more specifically adapted to the underlying biology, I suggest that you repeat the analyses for different groups of focal individuals: for juveniles and for reproductive trees, or at least for large trees ($dbh > 10$ cm) and small trees ($dbh < 10$ cm).

Lines 465-467

There is a potential issue with comparing the $\alpha_{ij,modified}$ and the $\alpha_{ij,true}$ coefficients. I expect that the quantities PAIR_{ij} and HOI_{ijh} (and PAIR_{ih} and HOI_{ihh}) should be highly correlated. This would be the case if the many third-order heterospecific neighbors k exert in their sum a largely uncorrelated stochastic effect on the second-order neighbors j of the focal individual i . Or in other words, some neighbors j exert somewhat stronger interactions to the focal tree i , but others somewhat weaker effects, but in the sum this stochastic modulations will average out (and the correlations appear). Therefore I want to see for each forest plot separately the correlation coefficients between these quantities.

In case of strong correlations between two quantities, you need to remove one quantities from the model because collinearity can cause problems with the interpretation of coefficients. You may then compare the pairwise model 2 (based on PAIR_{ii,modified} and PAIR_{ih,modified}) with a higher-order model based on the quantities PAIR_{ii,true}, PAIR_{ih,true}, HOI_{iii} and HOI_{ihh} (i.e., HOI_{ijh} and HOI_{ihh} are removed).

Additionally, I am not sure if it is conceptually justified to include both, the pairwise and the HOI terms into the statistical models. This is because the HOI terms refine the pairwise terms. In the pairwise model, the contributions $DBH_{ip} / d_{[i,ip]}$ of conspecific second-order neighbors i_p to neighborhood crowding (of the focal tree) are only weighted by the interaction coefficient α_{ii} , but in the expanded HOI model, they are weighted by the crowding indices of the second-order neighbors i_p times the coefficients β_{ijh} . The underlying idea is that a conspecific second-order neighbors i_p may compete less than α_{ii} if it is weakened by too strong crowding, or it may compete stronger than α_{ii} if its neighborhood is less crowded (the same applies for second-order heterospecific neighbors h_p). So, if the HOI terms are refinements of the pairwise terms, they should not appear both in a model.

lines 467-469

The inclusion of HOIs is statistically supported when the HOIs-inclusive models have AIC value that are at least a value of 2 smaller than the AICs of the competing models. If the AIC of two alternative models differs by less than two, you should select the simpler model.

lines 480-482

This sentence is unclear. Remember the reader that $\alpha_{ij, modified}$: pairwise coefficients estimated with (pairwise models) 2 and $\alpha_{ij, true}$: pairwise coefficients estimated with model 3 that also included HOI

lines 493-526

This is the weakest and most cryptic part of your analysis. I have the impression that the concepts of structural stability have not yet been sufficiently matured for the analyses you want to carry out, and propose to remove it.

I assume that you use the matrix of the $\alpha_{ij, modified}$ and $\alpha_{ij, true}$ as interaction matrix A . However, this approach is incorrect because of a scaling issue. You estimate the pairwise interaction coefficients and the HOIs at the plant neighborhood scale

(which is fine), but you cannot use them directly as interaction coefficients in a Lotka Volterra multispecies model because this model operates not at the neighborhood scale, but at the community scale. To derive the community scale interaction coefficients you need to upscale the coefficients from the neighborhood to the community scale (see reference 42). The coefficients at the community scale are emergent quantities that depend on the coefficients at the neighborhood scale, but also on spatial patterns such as intraspecific aggregation introduced via the pairwise and higher order interactions. In general the coefficients are not the same at these two scales. While the upscaling is resolved for pairwise interactions, further work is required to conduct a similar scale transition for higher-order interactions.

Figure S5.

Which α 's do you show here? Are this the "true" α 's in the third type of model (the one with HOIs), or are the α 's the "modified" ones from the second type of model (the one with only pairwise interactions). The difference between these two types of α 's must be made more clear since this is a key for understanding your approach.

(Remarks on code availability)

Referee #2

(Remarks to the Author)

The manuscript demonstrates that higher-order interactions are widespread among forest trees and diminish along latitude, and highlights that these interactions contribute to the latitudinal gradient in forest tree diversity. These findings provide insight for our understanding of the mechanisms underlying latitudinal diversity gradients. I have several comments and suggestions, outlined below.

The forest plots used in the study are distributed across the Northern Hemisphere, with none located in the Southern Hemisphere. I question whether it is appropriate to describe the findings as being on a "global scale" given this limitation. I recommend addressing this point and, at the very least, revising the wording accordingly.

L186-191 The findings reveal the latitudinal gradient in both competitive interactions and facilitative effects among conspecifics. Is there a specific reason why facilitative effects decline with latitude? Furthermore, I am curious about how to understand the simultaneous presence of significant gradients in both opposing interaction types—competition and facilitation—and their ecological implications.

P208-212 Given that higher-order interactions were not directly incorporated into the structural stability analysis, yet are highlighted as an important contribution to the diversity gradient, it would strengthen the manuscript to briefly explain how the authors used α true and α modified to indirectly account for HOIs before presenting the results.

L237-238 This part appears to be unexpected. It was not clearly explained in the Results or Methods sections. Instead, the Introduction highlights that the structural stability include "different types of biotic interactions (competitive or facilitative, pairwise or HOIs) simultaneously". The manuscript might benefit from a more detailed explanation of how these interaction types (HOIs) were simultaneously incorporated into the analysis—particularly in lines 208–212 or the corresponding Methods section.

Figures:

Figure 3 The abbreviations for the different types of higher-order interactions (title of y axis) are difficult to follow for readers. It would be helpful to include a simple schematic diagram within each panel to visually illustrate the relationships between the focal species, its neighbors, and the second-order neighbors (e.g., iii or iih).

Figure 3 Please carefully check the consistency between the figure and the figure legend. In lines 573–575, it is stated that "The species-level pairwise and higher-order interactions were related to latitude separately for competitive ($\alpha < 0, \beta < 0$, blue) and facilitative interactions ($\alpha > 0, \beta > 0$, orange) using exponential regressions." However, the colors mentioned here are not used to represent competitive or facilitative interactions in the figure.

Line to line comments:

Main

L195 Do you want to cite Figure S5 instead of S4? Please check.

L244 Is this sentence grammatically correct? It seems that "and" should be added after "intraspecific."

Methods

L395 Could you please clarify the meaning of "potential growth" and briefly explain how it was calculated?

L433 As I understand, there may be an issue with the last part of the formula. Should it be "PAIR_{ii} + PAIR_{ih}"? The same applies to equation (6).

L442-447, Could you please provide a more detailed explanation of the parameter m and how it is set to be species-specific?

L499 Could you please provide a clearer explanation of this equation? While the equation is presented, it would be helpful to offer a detailed explanation of the parameters involved and their meaning.

(Remarks on code availability)

The provided code is clearly structured. However, a key concern is that the most important parts of the analysis—namely, the calculation of interactions and the fitting of the growth and survival models—are demonstrated only using toy data rather than the original dataset. This limitation could affect the reproducibility of the study's main results.

Referee #3

(Remarks to the Author)

Higher-order biotic interactions enhance the global latitudinal tree diversity gradient

Yuanzhi Li et al.

The authors investigate the latitudinal gradient in species diversity (LDG). A gradient that has been known for a long time but for which the driver(s) remain poorly understood. Conspecific negative density dependence (CNDD) has been mentioned as a potential driver for this gradient but the authors suggest the effect has not been shown conclusively. This is somewhat debatable, as can be seen from references 14 and 15.

They show that Higher Order Interactions, interactions of other species on the effect between conspecifics and heterospecifics have a significant effect.

The use of HOI is new, for as far as I know. However, in some of the references used, the link between CNDD and the LDG has already been shown, even with data from the same ForestGeo Network.

The math is rather complicated and as the text is rather full with mathematical material it is not easy to read.

Finally, I missed some discussion on the hypothesis in line 157, and the fact that not only CNDD decreased with latitude but also positive density effects did.

Below are some questions and comments that I hope are useful to improve the manuscript.

Hans ter Steege

Line 118: This is not what ref 11 states, rather the opposite "The proportion of species affected is equivalent to that in tropical forests, failing to support the hypothesis that this mechanism is more prevalent at tropical latitudes"

Line 121: Here that is even acknowledged.

Line 122. Reference 14, actually shows a rather strong effect of CNDD on the LDG, based on plots of the same ForestGeo network.

Line 122: Ref 15, shows that richer forests in the US show more CNDD, a result also shown here.

Line 122. Hence, I am not so convinced that the findings of the LDG are less inconsistent with the results provided here. Of course, the testing of the HOI's is new.

Line 123: add gradient

Line 157: Therefore, we hypothesize that higher structural stability in the tropics supports the higher tree diversity – see below

Line 159: I am surprised that in South America the three richest forest plots (Yasuni, Amacayacu, and Manaus) have not been included. La Planada (Colombia) is a high montane forest.

Line 190. But apparently also does positive conspecific facilitation.

Line 191. So why conclude here that only CNDD is a driver of tree diversity, while the effect of CPDD is equally frequent and shows a similar latitudinal gradient?

Line 201. Is that not simply because in higher latitudes there are less species and thus more conspecifics in the surroundings?

Line 234. Fig. S2 is the first Suppl Fig called.

Line 234. This also does not support your hypothesis of line 157. Also it starts to drop off at very low species richness. What does that mean for your hypothesis?

Line 245. But they also strengthen them (Fig 3).

L395: How is potential growth determined?

L462: Using the log reduces the errors considerably.

(Remarks on code availability)

I am sorry, I did not test the code.

Version 1:

Reviewer comments:

Referee #1

(Remarks to the Author)

The new version of the manuscript improved substantially and the authors have done a great job of answering my questions and implementing changes! Overall, the study shows high originality and significance and is a long-needed contribution that may change the way species interactions in plant communities are conceptualized. The data are impressive and the methodology is valid, but needs a little more explication. The revision now has a clearer storyline and the methods gained substantially in clarity, as the text and the methods are now focussed on the heart of the analysis, with the robustness tests, structural stability, and the complicated-looking equations for the detailed calculations of the different crowding indices moved into the supplement. The new notation of the equations also helps. Nevertheless, I have a number of more minor questions and suggestions to strengthen the line of arguments of the manuscript. Additionally, a grammar check may be needed as there are several somewhat unusual phrases.

General comments

1) Thank you for re-running the analysis for large and small trees separately, it is good that the results for large trees are consistent with the results shown in the main text for all trees. You may include the analogous analysis to figure 5 for large trees to the supplement.

2) I am happy that you now examine the question (Q3) of how the cumulative effects of pairwise and higher-order interactions on growth (and survival) change with species abundance and latitude (instead of conducting A structural stability analysis). I found the new figure 5 that reports these results very interesting (i.e., a rare species advantage with respect to growth, which was stronger at lower latitudes).

3) Thank you for showing the correlations between the different pairwise and higher-order neighbourhood crowding indices. This figure should go into the supplement to allow the reader a better understanding of the properties of the higher-order indices. You may also mention that the correlation between $n_{i,jf}$ and $n_{ii,jf}$ arises because $n_{ii,jf}$ is approximately the square of $n_{i,jf}$, and the correlation between the heterospecific indices $n_{i,jf}$ and $n_{hh,jf}$ arise because $n_{hh,jf}$ is approximately the square of $n_{h,jf}$.

Ok, your argument is that you can retain both the pairwise and the HOI terms because this would be similar to using a linear term and a squared term in a regression. This should also be mentioned in the methods.

However, I would propose a somewhat different, more biological argument for retaining pairwise and HOI terms. First, comparing e.g., equations S3 and S8, you see that the higher-order crowding index $n_{ih,jf}$ weights each conspecific neighbour i_p by (i) its DBH, (ii) its distance $d_{[if,ip]}$ to the focal individual if , and (iii) by its heterospecific crowding index $n_{h,ip}$. Thus, the indices $n_{ih,jf}$ and $n_{ii,jf}$ refine the pairwise index $n_{i,jf}$ in the same way as weighting by DBH refines the crowding index that just counts neighbours and weights them by distance. Analogously, the indices $n_{hh,jf}$ and $n_{hi,jf}$ refine the index $n_{h,jf}$.

The argument of weighting (e.g., in case of competitive interactions) by $1/d_{[if,ip]}$ is that more distant neighbours compete less, and the argument of weighting by its DBH is that larger trees may compete more strongly. Now, this argumentation does not really work for the weighting with $n_{h,ip}$ because a conspecific neighbour i_p with a very large index $n_{h,ip}$ should show a weaker competitive ability, but not a larger one as suggested by the over-proportionally large weight $n_{i,ip}$. However, weighting by $(1 - \alpha n_{h,ip}/E[n_{h,ip}])$, instead of weighting by $n_{h,ip}$, does the job, which basically introduces the pairwise term! $E[n_{h,ip}]$ is the expectation of $n_{h,ip}$ for conspecifics i_p , and α is the strength of the higher-order effect. The term $\alpha/E[n_{h,ip}]$ is implicitly included in your regression coefficients α_{ii} and β_{iih} , but it might be interesting to calculate α . The weight $(1 - \alpha n_{h,ip}/E[n_{h,ip}])$ explains also the negative correlation between α_{ii} and β_{iih} and between α_{ih} and β_{ihh} .

Specific comments

line 115-117

Perhaps better "Theory suggests that (...) may greatly modify the pairwise interactions"

line 118-119

Perhaps: "However, there is a lack of empirical studies that investigate how HOIs intertwine with pairwise interactions, and how they may contribute to the latitudinal diversity gradient."

line 122-124

There is still a gap between the rare species advantage with respect to growth and survival and promotion of species diversity. So I suggest to be more conservative here, for example: "More importantly, HOIs were found to benefit rare species while disadvantaging common species, which suggests a mechanism to promote species diversity."

line 148

At this stage of the manuscript you propose the hypothesis that HOIs may have an important effect, I would therefore suggest to add "potential effects" to make this clearer: "(...) neglecting the potential effects of higher-order interactions (HOIs) that

(...)"

line 187

I would add: "on growth and survival": "(...) of pairwise interactions and HOIs on growth and survival (...)"

lines 193-194

What's about tree survival rates? Add the panel analogous to Fig. 3b to Extended Data Figure 3.

Extended Data Figures 1, 4

lines 11, 40, 41

There are typos in the subscripts, it must be: "(...) higher-order interaction coefficients β_{hi} and β_{ih}

lines 202-206, Figure 4

The result that some α and β values decrease with increasing latitude is an interesting pattern that apparently suggests that the average strength of true intraspecific pairwise interactions ($\alpha_{ii,true}$) and of HOIs would decline rapidly with latitude. Because of this catchy suggestion, it is even more important to ensure that this pattern is not the result of a systematic change in another variable across latitude.

Your scaling of the different indices $n_{i,if}$, $n_{ii,if}$, $n_{ih,if}$, and $n_{hi,if}$ with conspecific abundance N_i (equations S17 and S18) suggests that this result could potentially be caused by systematic changes in the values of the crowding indices with latitude. Note that $n_{h,if}$ and $n_{hh,if}$, which scale with heterospecific abundance N_h (that varies little compared to N_i), do not show the declining trend.

For example, the impact of the conspecific pairwise interactions on growth is $\exp(\alpha_{ii-ni,if})$ and $n_{i,if}$ scales with conspecific abundance N_i (eq. S17). Given that there are fewer species in forests at higher latitudes, species abundances will tend to be higher and therefore $n_{i,if}$ will also tend to be higher. It is therefore possible that systematic changes in $n_{i,if}$ with latitude may counteract the trend in α and β values and lead across latitude to similar range of the net effects of pairwise interactions on growth [i.e., $\exp(\alpha_{ii-ni,if})$]. To exclude this possibility and to verify that the net effect of conspecific interactions on growth (and perhaps survival) decline with latitude, you could expand Figure 5 to show the relative changes in growth rate along latitude due to the six different components $\exp(\alpha_{ii-ni,if})$, $\exp(\alpha_{ih-nh,if})$, $\exp(\beta_{iii-nii,if})$, $\exp(\beta_{iih-nih,if})$, etc. for two or three typical abundances.

line 213

add: "than by conspecific neighbors."

lines 226-238

I think this is a good idea to show in the main text how the pairwise and higher-order interactions modify the growth of individual trees on average across latitude to provide an answer to question Q3. The results are very interesting!

From lines 423-427 it follows that values of RCPAIRS and RCHOIs larger than one promote the growth on average (and values smaller than one reduces the growth on average). To facilitate interpretation of Figure 5, I would keep the logarithmic y-axis in Figure 5, but show instead of the log-transformed axis values (i.e., 4, 2, 0, -2 and -4) the corresponding untransformed values. With this small change, the reader can easily assess the magnitude of the average impact of the different interaction types.

For extended Data Figure 5 the interpretation becomes more complicated because survival does not scale linearly with the terms $\exp(\text{pair})$ and $\exp(\text{HOI})$. How did you calculate the relative changes in survival due to pairwise and HOI interactions?

lines 233-236

Not really, the effect of pairwise interactions is weaker (with the slope becoming non-significant for temperate forests), but not "relative constant". Using a linear scaling of the y-axis (i.e., the untransformed RCPAIRS and RCHOIs values) would show this clearer.

lines 236-238

Perhaps: "Taking together, our findings suggest that latitudinal changes in HOIs and pairwise interactions may strengthen the latitudinal tree diversity gradient by promoting growth of less abundant species at lower latitudes more strongly, with weaker contributions from pairwise interactions."

lines 422-424

To further assess how the latitudinal change of pairwise interactions and HOIs may contribute to the latitudinal tree diversity gradient (Q3), we first calculated the relative change in growth rate and survival probability caused by the neighbourhood (...)

Supplementary Text

lines 70-133 are missing.

(Remarks on code availability)

Referee #2

(Remarks to the Author)

The authors have done a great job addressing the previous concerns and comments raised. I only have some minor comments now.

L138-139 Part of the statement is wrong. Ref 11 (Daniel J. Johnson 2012), they found "species-rich regions exhibited stronger CNDD than species-poor regions" instead of "significant decline in CNDD with increasing latitude or species richness". Please revise it.

L227 Does "cumulative effects" refer to the combined influence of both competitive and facilitative effects? If so, please clarify this explicitly when the term is first introduced.

Method

L377 what is " λ_i "? Does it have any ecological meaning?

I wonder whether, in the scenario β_{ihh} , you also consider cases in which the heterospecific neighbour is a third species, rather than one of the two species involved in the focal pairwise interaction. My understanding is that this is the case; however, the notation " β_{ihh} " suggests otherwise and could be potentially confusing.

Supplementary text L70-134, please check, the content is incomplete.

(Remarks on code availability)

Referee #3

(Remarks to the Author)

I would like to thank the authors for a thorough rewrite. Although the math is still complicated to a general community ecologist like myself, I am convinced that the analyses and the text are strong.

Just a few things caught my eye when reading

line 234: should read 'relatively constant'

line 240-241: I would suggest "We would like to note some limitations of our analyses for future consideration, however." Other wise there is a focus on 'we'.

line 246: 'upscaled to the communities'

with kund regards

Hans ter Steege

(Remarks on code availability)

Version 2:

Reviewer comments:

Referee #1

(Remarks to the Author)

I congratulate the authors on their thorough revision of the manuscript and for carrying out all the additional analyses in response to my comments, which have clarified my remaining queries. The manuscript is an important contribution!

I only have a few minor comments left. I am fine with the absolute-modification form in equation (2) for interpreting the pairwise and HOI terms. However, it is important to briefly mention in the methods how the pairwise and HOI crowding indices are estimated (see below).

Specific comments

lines120-122

To be more specific, you may add that you analyze survival and growth of individual trees.

lines 149-150

This is true for the Lotka-Volterra type models that operate at the species level. However, in your study you consider survival and growth of individual trees based on neighborhood interactions. I would therefore suggest to change in according to figure 1 to “the potential effects of higher-order interactions (HOIs) that emerge when pairwise interactions between two neighbored trees are modified by other neighbors (Fig. 1).”

line 185-186

For better understanding add “, the true interaction between two trees of species i and j in the absence of other neighbors”: (...) whereas pairwise interactions estimated from HOI-inclusive models isolate HOIs and thus represent $\alpha_{ij,true}$ (orange arrows in Fig. 1), the true interaction between two trees of species i and j in the absence of other neighbors.

Figure 5

It would be interesting to see also the relative change in growth rate caused by the cumulative effects of the modified pairwise interactions (i.e., based on the PAIR-only model).

line 241

You could start a new paragraph with “Taken together”.

line 412

It is important to give the reader an idea of how the crowding indices and the higher order crowding indices are estimated. You may add before “The calculation (...)”:

“In short, the conspecific and heterospecific pairwise neighborhood crowding indices $n_{i,if}$ and $n_{h,ih}$ add up the crowding contributions of all conspecific and heterospecific neighbors of a focal individual within a given radius, respectively. The contribution of a neighbor is an increasing function of the dbh of the neighbor and a decreasing function of the distance to the focal individual (eq. S3, S4). The contribution of a neighbor in the corresponding higher-order crowding indices $n_{ii,if}$ and $n_{ih,if}$ is additionally multiplied by its conspecific crowding index (eq. S7, S9), and that of $n_{ih,if}$ and $n_{hh,if}$ is additionally multiplied by its heterospecific crowding index (equations S8, S10).”

line 441

add “true”: “true pairwise interactions”

(Remarks on code availability)

Dear [REDACTED],

We first want to thank the reviewers for their constructive and insightful feedback which greatly improved our study. We also thank you for inviting resubmission. We have carefully read and thoroughly addressed the comments and criticisms of the three reviewers. Among the many changes we have made, some of the main changes include:

- (1) Following the suggestion of reviewer #3, we attempted through various channels but were eventually not able to acquire the three forest plots data from Central and South America (Yasuni, Amacayacu, and Manaus). Instead, we added a plot from West Africa (Rabi plot, Gabon), which did not affect our main results and conclusions. To reflect the omission of forests from South America, we have now replaced the word “global” with “the Northern Hemisphere” or “Northern Hemisphere forests”, so that to provide a more precise description as suggested by reviewer #2.
- (2) We substantially revised the method section to improve clarity and readability. Specifically, we used new notations to represent pairwise (n_{i,i_f} , n_{h,i_f}) and higher-order (n_{ii,i_f} , n_{ih,i_f} , n_{hi,i_f} , n_{hh,i_f}) neighborhood crowdedness around the focal tree i_f , and moved the detailed calculations into Supplementary Text 1. In addition, we moved all robustness tests and related results (including choice of parameters, spatial autocorrelation, uncertainty of interaction coefficients, tree size) into Supplementary Text 2, and cited them properly in the main text.
- (3) We conducted two additional analyses to address the second major comment of Reviewer #1. They are: (1) examining how cumulative effects of HOIs change with species abundance and latitude, and (2) upscaling interaction coefficients and applying them to structural stability analyses. All the materials related to upscaling of the coefficients and structural stability were presented in Supplementary Text 3.

We have provided a detailed, point-by-point response to the reviewers’ comments. We hope you agree that our revision is thorough sufficient. The study is substantially improved. We look forward to further feedback from you and the reviewers.

To ease referring and reading, we highlighted our changes in blue in the revised manuscript.

Best regards,

Chengjin Chu and co-authors

Referee #1 (theoretical ecology, modeling)

This study uses a large data set of fully mapped forest inventory plots around the globe to study the effects of pairwise and higher-order interactions on growth and survival of individual trees. The objectives are to assess if higher-order interactions play globally an important role in governing growth and survival in forests, if pairwise and higher-order interactions change systematically along latitude, and if latitudinal gradients in higher-order interactions would be related to the latitudinal gradient in tree diversity. The study is a long-needed contribution that may change the way species interactions in plant communities are conceptualized. The authors compiled an impressive data set to address their objectives. However, while I see a large potential in the study, the current analyses and descriptions go often not deep enough and need some revision. I have three major comments and a number of smaller comments and suggestions to improve the work.

Response: Thank you for your encouraging comments and constructive suggestions to improve our study. We have carefully read and addressed your concerns in the revision. Please see our explanations below how we incorporated your comments and criticisms in the revised study. To avoid redundancy of response, we group questions of similar comments and respond to them together.

1) To improve the biological interpretation of the analyses I propose that the authors conduct separate analyses for juveniles and for reproductive trees or for large trees ($\text{dbh} \geq 10\text{cm}$) and small trees ($\text{dbh} < 10\text{cm}$). Because the vast majority of all trees in your analysis will be very small trees your current results are heavily driven by small trees, but for demographic assessment you need to deal with larger reproductive trees. In any case, it would be interesting if the HOI (and pairwise) effects change with tree size.

lines 160-163

The aims are somewhat unspecific with respect to the underlying biology. I guess that you conduct your estimates for all focal individuals in the forest plots, without considering size or

reproductive status. Given that the vast majority of all trees will be very small trees (say dbh < 2.5 cm) your results will be heavily driven by small trees, but a demographic Lotka-Volterra model (used to derive structural stability) deals with reproductive trees. So, I like to see separate analyses for juveniles and for reproductive trees or at least for large trees (dbh \geq 10cm) and small trees (dbh <10cm).

lines 180-181

This interesting result is driven by small trees, what happens for larger trees?

Lines 458-461

Given that the vast majority of all trees you analyzed will be very small trees (say dbh < 2.5 cm) your results will be heavily driven by small trees if you use all individuals together. However, small saplings and adult trees may behave quite differently. To make your analysis more specifically adapted to the underlying biology, I suggest that you repeat the analyses for different groups of focal individuals: for juveniles and for reproductive trees, or at least for large trees (dbh $> 1=10$ cm) and small trees (dbh <10cm).

Response: Thank you for this important and constructive suggestion. We agree that the sign and strength of biotic interactions can be dependent on tree size, and that separating small versus large trees could provide clearer biological interpretation. Following your suggestion, we repeated the analyses separately for small trees ($DBH < 10$ cm) and large trees ($DBH \geq 10$ cm). We reported the results in Supplementary Text 2.4, which show: (i) HOIs are prevalent in both large and small trees; (ii) pairwise and higher-order interactions (except β_{ihh}) on small and large trees also decline with latitude, consistent with the overall pattern of when all trees combined.

Given the consistency across size classes, we retain the results that do not separate small and large trees in the main text but point out that the results are robust to tree size classes (lines 432-436).

2) The section on structural stability is the weakest and most cryptic part of your analysis. While I understand that it is tempting to investigate the consequences of the interesting patterns found, it leaves the impression that the concepts have not yet been sufficiently matured for the analyses you want to carry out. I therefore propose to remove it and replace it by a deeper biological analysis of pairwise and HOT interactions, e.g., by investigating how the importance of HOI and pairwise effects change with tree size. Regarding the structural stability analysis, it is incorrect to use the interaction coefficients of a neighborhood-scale analysis in a community-scale demographic model, you need to conduct an additional upscaling step to translate the interactions at the scale of individual trees to Lotka-Volterra interaction coefficients.

lines 172-174

There is a scaling issue when you apply your interaction coefficients to structural stability. You estimate the pairwise interaction coefficients and the HOIs at the plant neighborhood scale, but to get the appropriate interaction coefficients at the community scale to be fed into Lotka-Volterra models you need to upscale (see reference 42).

lines 493-526

This is the weakest and most cryptic part of your analysis. I have the impression that the concepts of structural stability have not yet been sufficiently matured for the analyses you want to carry out, and propose to remove it. I assume that you use the matrix of the α_{ij} modified and $\alpha_{ij,true}$ as interaction matrix A . However, this approach is incorrect because of a scaling issue. You estimate the pairwise interaction coefficients and the HOIs at the plant neighborhood scale (which is fine), but you cannot use them directly as interaction coefficients in a Lotka Volterra multispecies model because this model operates not at the neighborhood scale, but at the community scale. To derive the community scale interaction coefficients you need to upscale the coefficients from the neighborhood to the community scale (see reference 42). The coefficients at the community scale are emergent quantities that depend on the coefficients at the neighborhood scale, but also on spatial patterns such as intraspecific aggregation introduced via the pairwise and higher order interactions. In general, the coefficients are not the same at these two scales. While the upscaling is resolved for pairwise interactions, further work is required to conduct a similar scale transition for higher-order interactions.

Response: We appreciate very much your insightful comments and constructive suggestions. We agree that our structural stability analysis, as originally presented, relied on scale transition from neighborhood-level interactions to community-level interactions. Although it is challenging to upscale the neighborhood interactions to assess contribution of HOIs to the latitudinal diversity gradient, we spare no effort to explore the link between our observed latitudinal pattern of HOIs and latitudinal tree diversity gradient. Following your stimulating suggestion, here we offer two possible solutions.

(1) Examining how cumulative effects of HOIs change with species abundance and latitude

We calculated the relative changes in growth rate and survival probability caused by cumulative effects of pairwise interactions and HOIs individually, and then examined how these effects changed with species abundance and latitude. We found that the relative changes in growth rate caused by cumulative effects of pairwise interactions and HOIs both declined with species abundance, shifting from positive for rare species to negative for common species (Table 1 and Fig. 5). It implies that cumulative effects of pairwise interactions and HOIs promote growth rate of rare species but suppress that of common species, thereby potentially maintaining species diversity. Moreover, the stabilizing effects of pairwise interactions remained relatively constant across latitudes (Fig. 5), whereas the stabilizing effects of HOIs became weaker toward higher latitudes (Fig. 5). Therefore, latitudinal changes of HOIs could enhance the latitudinal tree diversity gradient, while latitudinal changes of pairwise interactions contribute little to this gradient. We have added the new analyses and results in the main text (lines 422-429 and 222-238).

(2) Upscaling interaction coefficients and applying them to structural stability analyses

Thanks to the generous help from Prof. Thorsten Wiegand, we upscaled the individual-level interaction coefficients to population-level interaction coefficients following Wiegand et al. (2021; 2025), and then applied the population-level interaction coefficients to structural stability analyses. We found a marginally positive relationship between SES of structural stability (based on α_{true} and $\alpha_{modified}$) and species richness (Supplementary Information Fig. S9). However, we like to note that our result should be treated as an approximation because there is no analytical

solution to upscaling the HOI coefficients, nor including them into structural stability. To avoid overinterpretation, we moved all contents related to the upscaling and structural stability into Supplementary Text 3, where they are presented as complementary and exploratory analyses.

Although neither solution above offers an exact answer to the question of how HOIs contribute to latitudinal tree diversity gradient (Q3), they represent a concrete step closer to the solution. We decide to retain this section but be clear about the limitations of our current study (lines 241-250).

3) I like to see a correlation analysis of the quantities PAIR_{ii} and HOI_{iih} and the quantities PAIR_{ih} and HOI_{ihh}. If they are strongly correlated (as I suspect), one of each must be removed from the model. Conceptually, I doubt that it is correct to include both pairwise and HOI interactions in the survival and growth models. My argument is that the HOIs basically refine the pairwise interactions by allowing a neighbor to compete more or less than the average (assumed by pairwise interactions), depending on the crowding it suffers itself.

Line 194-202

The correlations between α_{ii} and β_{iih} and α_{ih} and β_{ihh} are quite high. I am not entirely convinced by your interpretation. Did you check if the different variables in your model are correlated? With equations 5 and 6 you have six interaction terms in your equations 1 and 2, I guess that the terms PAIR_{ii} and HOI_{iih} as well as the terms PAIR_{ih} and HOI_{ihh} would be strongly correlated. If pairs of PAIR and HOI terms are strongly correlated, you need to remove one of them from the model.

Lines 465-467

There is a potential issue with comparing the $\alpha_{ij,modified}$ and the $\alpha_{ij,true}$ coefficients. I expect that the quantities PAIR_{ii} and HOI_{iih} (and PAIR_{ih} and HOI_{ihh}) should be highly correlated. This would be the case if the many third-order heterospecific neighbors k exert in their sum a largely uncorrelated stochastic effect on the second-order neighbors j of the focal individual i . Or in other words, some neighbors j exert somewhat stronger interactions to the

focal tree i , but others somewhat weaker effects, but in the sum this stochastic modulations will average out (and the correlations appear). Therefore I want to see for each forest plot separately the correlation coefficients between these quantities.

In case of strong correlations between two quantities, you need to remove one quantities from the model because co-linearity can cause problems with the interpretation of coefficients. You may then compare the pairwise model 2 (based on PAIR_ii,modified and PAIR_ih,modified) with a higher-order model based on the quantities PAIR_ii,true, PAIR_ih,true, HOI_iii and HOI_ihi (i.e., HOI_iih and HOI_ihh are removed).

Additionally, I am not sure if it is conceptually justified to include both, the pairwise and the HOI terms into the statistical models. This is because the HOI terms refine the pairwise terms. In the pairwise model, the contributions $DBH_{ip} / d_{[if,ip]}$ of conspecific second-order neighbors ip to neighborhood crowding (of the focal tree) are only weighted by the interaction coefficient α_{ii} , but in the expanded HOI model, they are weighted by the crowding indices of the second-order neighbors ip times the coefficients β_{iih} . The underlying idea is that a conspecific second-order neighbors ip may compete less than α_{ii} if it is weakened by too strong crowding, or it may compete stronger than α_{ii} if its neighborhood is less crowded (the same applies for second-order heterospecific neighbors hp). So, if the HOI terms are refinements of the pairwise terms, they should not appear both in a model.

Response: Thank you for raising the collinearity issue. To avoid confusion, we first introduced new notation in the revision. The terms n_{i,i_f} and n_{h,i_f} refer to conspecific and heterospecific neighborhood crowding indices, respectively. The higher-order neighborhood crowding indices n_{ii,i_f} , n_{ih,i_f} , n_{hi,i_f} and n_{hh,i_f} denote intraspecific pairwise interactions modified by conspecific neighbors ($\alpha_{ii} \leftarrow i$), intraspecific pairwise interactions modified by heterospecific neighbors ($\alpha_{ii} \leftarrow h$), interspecific pairwise interactions modified by conspecific neighbors ($\alpha_{ih} \leftarrow i$), and interspecific pairwise interactions modified by heterospecific neighbors ($\alpha_{ih} \leftarrow h$), respectively. Full details of their calculation are provided in Supplementary Text 1.

We then examined the correlations among the pairwise (n_{i,i_f} and n_{h,i_f}) and higher-order neighborhood crowding indices (n_{ii,i_f} , n_{ih,i_f} , n_{hi,i_f} and n_{hh,i_f}). As you rightly expected, some variables were highly correlated (see Fig. 1 below). Specifically, n_{i,i_f} is highly correlated to n_{ii,i_f} , while n_{h,i_f} is highly correlated to n_{hh,i_f} . However, the correlations between the variables do not necessarily translate into correlations between the corresponding coefficients. For example, α_{ii} was more strongly correlated with β_{ihn} than with β_{iii} (Extended Data Fig. 2), suggesting that intraspecific pairwise effects be more strongly modified by heterospecific neighbors than conspecific neighbors.

We agree that in a purely linear regression framework ($Y = c + \alpha X_1 + \beta X_2 + \varepsilon$), high collinearity among predictors (e.g., X_1 and X_2) would typically require removing redundant variables. However, our models are quadratic-like after transformation: $Y = c + \alpha X + \beta X^2 + \varepsilon$. Y is growth rate (or survival probability), X are neighborhood crowding indices (e.g., n_{i,i_f} and n_{h,i_f}), X^2 are higher-order neighborhood crowding indices (e.g. n_{ii,i_f} is approximate square of n_{i,i_f} ; n_{hh,i_f} is approximate square of n_{h,i_f}). We conducted a simple simulation for this quadratic relationship, showing that the correlations between estimated coefficients (α and β) can be weak or absent, even X and X^2 are highly correlated (see Fig. 2 below). It confirms that correlations among predictors do not necessarily translate into correlations among the estimated α and β coefficients. This has also been demonstrated in previous HOI studies (Mayfield & Stouffer 2017; Buche et al. 2024), which retained both pairwise and HOI terms despite strong correlations among them.

We tried to understand your conceptual justification. Yes, the pairwise effect of neighbor i_p on focal tree i_f in the absence of other neighbors is the contribution of i_p to neighborhood crowding weighted by intraspecific pairwise interaction:

$$\alpha_{ii,true} \frac{DBH_{i_p}^u}{d[i_f, i_p]^v}$$

When there are neighbors around i_p , the intraspecific pairwise interaction $\alpha_{ii,true}$ could be modified by conspecific neighbors and heterospecific neighbors of i_p , and the strength of modification depends on conspecific and heterospecific neighbor crowding of i_p (n_{i,i_p} and n_{h,i_p}):

$$n_{i,i_p} = \sum_{q=1}^{N_{i,i_p}} \frac{DBH_{i_q}^u}{d[i_p, i_q]^v}; n_{h,i_p} = \sum_{q=1}^{N_{h,i_p}} \frac{DBH_{i_q}^u}{d[i_p, i_q]^v}$$

Therefore, the modified pairwise effect of i_p on i_f in the presence of its neighbors is

$$\begin{aligned} \alpha_{ii,modified} \frac{DBH_{i_p}^u}{d[i_f, i_p]^v} &= \left(\alpha_{ii,true} + \beta_{iii}n_{i,i_p} + \beta_{iih}n_{h,i_p} \right) \frac{DBH_{i_p}^u}{d[i_f, i_p]^v} \\ &= \alpha_{ii,true} \frac{DBH_{i_p}^u}{d[i_f, i_p]^v} + \beta_{iii}n_{i,i_p} \frac{DBH_{i_p}^u}{d[i_f, i_p]^v} + \beta_{iih}n_{h,i_p} \frac{DBH_{i_p}^u}{d[i_f, i_p]^v} \end{aligned}$$

Both pairwise and higher-order interactions (α and β) are included in above equation to estimate their overall effects of its neighbor and neighbor's neighbors. Our interpretation of this mathematical formulation is that it is necessary to include both pairwise and HOI interactions.

We could misunderstand your point and would appreciate your further insights.

Fig. 1. Correlation matrix between pairwise (n_{i,i_f} and n_{h,i_f}) and higher-order (n_{ii,i_f} , n_{ih,i_f} , n_{hi,i_f} and n_{hh,i_f}) neighborhood crowding indices.

Fig. 2. Test the dependency of correlation between α and β on correlation between X and X^2 . In this simulation, we assumed a quadratic relationship between variable Y and X for 20 species ($Y = c + \alpha X + \beta X^2 + \varepsilon$). Each species was allowed to have different α and β . The correlation between α and β is determined by the p value: $\beta = -0.1p\alpha + (1 - p)\varepsilon$, $\varepsilon \sim U(0, 0.1)$. Higher p value implies higher correlation between α and β . The simulated correlation between α and β depends on their true correlation (p value, left panel), but not on the correlation between variables X and X^2 (right panel).

Specific comments

line 131-133

To estimate $\alpha_{ij,true}$, did you select only pairs of individuals without other neighbors, as suggested by the text: “the effect of a neighbor tree j on a target tree i in the absence of other trees”? Or is $\alpha_{ij,true}$ the coefficient determined in the analysis that includes both pairwise and HOI effects? I notice that you briefly define $\alpha_{ij,true}$ and $\alpha_{ij,modified}$ in lines 168-170, but this terminology must be made much clearer through the text.

Response: Sorry for not making that clear in the text. Ideally, $\alpha_{ij,true}$ should be estimated from individuals whose neighbors have no other neighbors (i.e., no interaction modification).

However, rarely can we find trees that are free of neighborhood interaction modification in our study forests (or any forest). Instead, we estimated $\alpha_{ij,true}$ from demographic models that explicitly included both pairwise and higher-order interactions from all focal individuals. By explicitly accounting for higher-order interactions, we can isolate the effects of true pairwise interactions from effects of higher-order interactions on demography. In contrast, when higher-order interactions are not considered, the pairwise coefficients implicitly incorporate effect of HOIs and thus denoted as $\alpha_{ij,modified}$. We have revised the description to make it clearer and more consistent in the notation (lines 181-185, 408-411).

Figure 1

I think panels b and c are slightly misleading since the presence of neighbors k of neighbor j modifies the interaction strength of neighbor j on the focal individual i. So, the arrows should go from k to j.

Response: As you may have noticed that Figure 1 is modified from a previous study (Levine et al. 2017). In that paper, the arrow from *k* directly to the arrow from *j* to *i* represents a higher-order interaction (interaction modification), whereas the arrow from *k* to *j* represents an interaction chain (another type of indirect interaction). Therefore, we prefer to retain the current illustration to be consistent with Levine et al. (2017).

line 133-136

This description does not really capture what you analyze here in forest systems where one focal tree has usually many neighbors. Alternatively, you could explain that the pairwise approach assumes that the performance (growth, survival) of a focal tree depends on the neighborhood crowding surrounding the focal individual *i* (that sums up the contribution of each con- and heterospecific neighbor *i_p* and *i_h* within a given distance *R* of the focal individual). However, the HOI approach assumes that the contribution of the neighbors *i_p* and *i_h* may be somewhat larger or smaller than assumed by the pairwise approach if one considers the crowding indices of the neighbors *i_p* and *i_h*. In case of competitive interactions, for example, a neighbor *i_p* may

compete more weakly if it suffers large than average neighborhood crowding, and it may compete more strongly if it enjoys a smaller than average neighborhood crowding.

Response: We accept the criticism that Figure 1 and its description do not capture the complexity of the forests, where a focal individual typically has many neighbors. We took the three simple diagrams to illustrate the concept of HOIs and how they can modify pairwise interactions. Following your comment, we added a new figure (Fig. S1) in Supplementary Information that depicts a focal tree surrounded by many neighbors and explained how the pairwise and higher-order crowding indices (Supplementary Text 1) are calculated. Hope this addition helps reader understand how our analyses are conducted.

line 138

Here you should cite the original study and provide the Latin names of the species. Note that the term “plantain” refers also to cooking banana, which is probably not what you mean.

Response: Thank you. We replaced this example with tree species whose Latin names were provided (lines 157-159).

lines 146-148

You should add here that one should first investigate the effect of HOIs on plant performance (e.g., growth and survival) before targeting coexistence.

Response: One sentence is now added to state the effects of HOIs on tree performance (line 160) and related references are also cited (Li et al. 2021, Lai et al. 2022).

Line 148

Perhaps better: “the recently developed structural approach for understanding multispecies coexistence”

Response: We have revised the sentence following your suggestion (line 107 in Supplementary Information) and move all content related to structural stability into Supplementary Text 3.

lines 163-172

This is a nice approach, however, did you consider models with $\Delta_{AIC} < 2$ to be equally supported by the data? You mention only that you select the model with the lowest AIC. Please clarify. If the AIC of pairwise and HOI differ by less than two, you should select the simpler model (the pairwise one).

lines 176-178

If the AIC of pairwise and HOI models differ by less than two, you should select the simpler model (the pairwise one). Here the reader needs to get an idea of how much better the HOI-inclusive models are in predicting growth and survival compared to the next best model. For growth you could show observed vs. predicted plots.

lines 467-469

The inclusion of HOIs is statistically supported when the HOIs-inclusive models have AIC value that are at least a value of 2 smaller than the AICs of the competing models. If the AIC of two alternative models differs by less than two, you should select the simpler model.

Response: Thank you for your suggestion of model selection. In the previous version, we simply selected HOIs-inclusive models with the lowest AIC without considering the AIC difference. Following your suggestion, we now only retain HOIs-inclusive models when their AIC values are at least 2 units smaller than other two alternative models (line 413). With this stricter condition, the proportion of the selected HOIs-inclusive models as the best models is only slightly reduced compared to our previous results (lines 190-193 and Fig. 3a). However, this does not at all change our conclusion about the prevalence of HOIs. Following your suggestion, we added a figure showing the correlation between observed and predicted growth rates (Fig.

3b). The HOIs-inclusive models on average increased 5~10% of correlation compared to the two alternative models.

line184-191

I am not sure if you can interpret positive coefficients generally as facilitation. Especially juvenile trees are often clustered together in smaller multispecies clumps e.g., caused by canopy gaps or because they are dispersed together by animals. If these sites are favorable microsites (where saplings grow and survive better than in the surrounding), this may also result in positive coefficients.

Response: We agree positive coefficients can arise from many factors (e.g., clumped dispersal or favorable microsites) other than facilitation between trees. To admit, we can never be certain or verify that positive coefficients represent genuine facilitation with the data in hands. However, the growth models including only small trees and 20×20 m quadrats as random effect show they have little effect on the estimated of interaction coefficients and do not change our conclusions. While we cannot completely rule out the influence of clumped dispersal or favorable microsites, the robustness of our results suggests that facilitation among neighboring trees is a more likely explanation for the observed positive coefficients.

Fig. 3.

The decline in the average positive and negative coefficients with latitude is an interesting pattern!

Response: Indeed, before we started the study, it was only our hope to find some kind of latitudinal patterns. We were over the moon to discover such a strong pattern. We believe our work is an important step toward mechanistic understanding of the latitudinal diversity gradients and toward reconciling the previous studies that focus on assessing the contributions of negative pairwise interactions (e.g. CNDD) to latitudinal tree diversity gradient (LaManna et al. 2017, Hülsmann et al. 2021, Hülsmann et al. 2024).

Lines 411

To help the reader, I would simplify the equations by setting $u = 2$ and $v = 1$ and $m = 0$ (similar to of line 456), but mention at an appropriate place that you tested additional scaling parameters u and v and saturation parameters m . I would use $u = 2$, because this means that species interactions would be mediated by basal area (which scales with biomass). Additionally, to be consistent with the established terminology I would suggest that you call the quantities $PAIR_{if}$ neighborhood crowding indices (or neighborhood competition indices) and cite one of the studies that developed this approach (e.g., Canham et al., 2004: doi: 10.1139/X03-232; Uriarte et al., 2004: doi: 10.1111/j.0022-0477.2004.00867.x). It may also help to express equation 6 in terms of the neighborhood crowding indices of the con- and heterospecific second-order neighbors i_p and h_p of the focal tree i_f .

Response: Thank you for the suggestions. We now use a set of new notations referring pairwise (n_{i,i_f} , n_{h,i_f}) and higher-order (n_{ii,i_f} , n_{ih,i_f} , n_{hi,i_f} , n_{hh,i_f}) neighbor crowding around the focal tree i_f , and have moved the details of their calculation into Supplementary Text 1. The cumulative effects of pairwise interactions ($PAIR_{i_f}$) and HOIs (HOI_{i_f}) will be the sum of the product of interaction coefficients and neighbor crowding indices (lines 381-397). We also move the materials about the choices of parameters into Supplementary Text 2.1, and retain the results in the case of $u=1$ in the main text because of lowest AIC compared to other cases.

Lines 427-432

The grouping into con- and heterospecifics is usually called “mean field approximation” (e.g., O’Dwyer & Chisholm 2014: Ecol. Lett. 17, 961–969) and rests on the assumption of diffuse competition, where the focal species interacts with an average taken over other species in the community. It rests on the diversity and variability of the local surroundings of each individual.

Lines 435-442

By grouping species into con- and heterospecifics the initial idea of species-specific higher-order interactions is somewhat lost. You should mention the con- vs heterospecific grouping earlier on in the main text to avoid wrong expectations of readers.’

Response: Thanks for these explanations. We removed the species-specific equations and added a sentence stating this mean-field approach and cited the suggested literature (lines 385-387).

lines 480-482

This sentence is unclear. Remember the reader that α_{ij} , modified: pairwise coefficients estimated with (pairwise models) 2 and α_{ij} , true: pairwise coefficients estimated with model 3 that also included HOI

Response: Following your suggestion, we have substantially revised the method section to improve the clarity and readability. This particular sentence is now removed. We also clearly explain α_{true} and $\alpha_{modified}$ (lines 182-185 and 408-411).

Figure S5.

Which α 's do you show here? Are these the “true” α 's in the third type of model (the one with HOIs), or are the α 's the “modified” ones from the second type of model (the one with only pairwise interactions). The difference between these two types of α 's must be made more clear since this is a key for understanding your approach.

Response: Thank you for asking for clarification. Original Figure S5 (now Extended Data Fig. 2) is intended to identify whether intraspecific pairwise interactions (α_{ii}) are more likely modified by conspecific (β_{iii}) or heterospecific (β_{iih}) neighbors, and whether interspecific pairwise interactions (α_{ih}) are more likely modified by conspecific (β_{ihi}) or heterospecific (β_{ihh}) neighbors. Therefore, this figure presents the correlations between α_{true} (rather than $\alpha_{modified}$) estimated from the third type of models and the higher-order interaction coefficients. We now add a sentence to clarify this point in legend of Extended Data Fig. 2.

Referee #2 (tropical forest ecology):

The manuscript demonstrates that higher-order interactions are widespread among forest trees and diminish along latitude, and highlights that these interactions contribute to the latitudinal gradient in forest tree diversity. These findings provide insight for our understanding of the mechanisms underlying latitudinal diversity gradients. I have several comments and suggestions, outlined below.

Response: Thank you for your insightful comments which greatly helped improve the study and presentation. Please see below our explanations how we addressed your comments in the revision. To avoid redundancy of response, we grouped questions of similar comments and responded to them together.

The forest plots used in the study are distributed across the Northern Hemisphere, with none located in the Southern Hemisphere. I question whether it is appropriate to describe the findings as being on a "global scale" given this limitation. I recommend addressing this point and, at the very least, revising the wording accordingly.

Response: This is rather a legitimate concern. As reflected by the long list of authorship, many colleagues have helped contribute their invaluable data to this study. Following reviewer #3's suggestion, in this revision we have attempted to expand the geographic coverage of study plots, particularly the plots from South America. We eventually were only able to add Rabi plot from Gabon in West Africa, which is also the only site located in the Southern Hemisphere. We tried all our resources to contact the PIs of plots of Yasuni, Amacayacu, and Manaus, but received no response. With the 32 plots, we agree that it is more appropriate to tune down the "global scale". In the revision, we adopt your suggestion to present our study as a "Northern Hemisphere" study, rather than a global one, and the rewording in the revision reflects that change (lines 121, 197, 253). Hope this change of presentation more accurately reflects the geographic scope of the study while preserving the broad implications of our findings.

L186-191 The findings reveal the latitudinal gradient in both competitive interactions and facilitative effects among conspecifics. Is there a specific reason why facilitative effects decline with latitude? Furthermore, I am curious about how to understand the simultaneous presence of significant gradients in both opposing interaction types—competition and facilitation—and their ecological implications.

Response: That is a great question. The latitudinal decline of facilitation has rarely been reported in previous studies because they primarily focus on competition. One possible reason for this pattern is that more species at low latitudes are associated with mutualists (e.g., AM mycorrhizal fungi, pollinators; see correlative evidence in Wiegand et al. 2025 Nature 640:967-973). We recognize it is much more challenging to pin down the causal effects, if ever possible. In the revision, we estimated the relative change in growth rate and survival probability caused by cumulative effects (including both competitive and facilitative effects) of pairwise interactions (RC_{PAIR}) and HOIs (RC_{HOI}) individually. We found that RC_{PAIR} and RC_{HOI} declined with species abundance, shifting from positive for rare species to negative for common species (Table 1 and Fig. 5). This finding implies that cumulative effects of pairwise interactions and HOIs promote rare species but suppress common species, thereby promoting species diversity. Moreover, the stabilizing effects caused by pairwise interactions remain relative constant across latitudes (Table 1 and Fig. 5), whereas the effects caused by HOIs become weaker towards higher latitudes (Table 1 and Fig. 5). These results suggest that HOIs play a primary role in enhancing the latitudinal diversity gradient, while the effect of pairwise interactions on this gradient is minimal. An important implication of the finding is that the competition-based classical theory is insufficient to explain the macroecological patterns of forest communities. The interactions of forest trees are much more complex and intricacy than what pairwise interaction can describe. High-order interactions matter in structuring forest communities.

P208-212 Given that higher-order interactions were not directly incorporated into the structural stability analysis, yet are highlighted as an important contribution to the diversity gradient, it would strengthen the manuscript to briefly explain how the authors used α_{ture} and α_{modified} to indirectly account for HOIs before presenting the results.

L237-238 This part appears to be unexpected. It was not clearly explained in the Results or Methods sections. Instead, the Introduction highlights that the structural stability include “different types of biotic interactions (competitive or facilitative, pairwise or HOIs) simultaneously”. The manuscript might benefit from a more detailed explanation of how these interaction types (HOIs) were simultaneously incorporated into the analysis—particularly in lines 208–212 or the corresponding Methods section.

Response: In the revision, we improved the description by more clearly explaining α_{true} and $\alpha_{modified}$, and how they were estimated (lines 408-411). In addition, following the suggestion of Reviewer #1, we moved all materials related to structural stability to Supplementary Text 3, where we showed step by step how these different interaction types were incorporated into structural stability.

Figures:

Figure 3 The abbreviations for the different types of higher-order interactions (title of y axis) are difficult to follow for readers. It would be helpful to include a simple schematic diagram within each panel to visually illustrate the relationships between the focal species, its neighbors, and the second-order neighbors (e.g., iii or iih).

Response: We have added a small schematic diagram modified from Figure 1 to illustrate the different types of interactions in each panel. It is now Figure 4 in the revision.

Figure 3 Please carefully check the consistency between the figure and the figure legend. In lines 573–575, it is stated that "The species-level pairwise and higher-order interactions were related to latitude separately for competitions ($\alpha < 0, \beta < 0$, blue) and facilitations ($\alpha > 0, \beta > 0$, orange) using exponential regressions." However, the colors mentioned here are not used to represent competitive or facilitative interactions in the figure.

Response: Thanks for pointing out this error. We originally distinguished competition and facilitation by color. Later, we decided to distinguish α_{true} and $\alpha_{modified}$ by color, consistent with Figure 1, but forgot to change the text in the legend. Corrected now (lines 493-494).

Line to line comments:

Main

L195 Do you want to cite Figure S5 instead of S4? Please check.

Response: Thanks for pointing out the typo. In the revision, we have reorganized all the tables and figures in Extended Data and Supplementary Information and carefully checked all citations.

L244 Is this sentence grammatically correct? It seems that "and" should be added after "intraspecific."

Response: Corrected now (line 253). Thank you.

Methods

L395 Could you please clarify the meaning of "potential growth" and briefly explain how it was calculated?

Response: The potential growth rate is growth rate in absence of neighbors, which are estimated from the growth model (the intercept). Now explained (line 373).

L433 As I understand, there may be an issue with the last part of the formula. Should it be "PAIR_{ii} + PAIR_{ih}"? The same applies to equation (6).

Response: Sorry for causing the confusion. We have now used a different set of notations (e.g. n_{i,i_f} , n_{h,i_f}) to refer neighbor crowding indices. The cumulative effects ($PAIR_{i_f}$) are the product

of interaction coefficients (e.g. α_{ii} , α_{ih}) and neighbor crowding indices (lines 381-397). To improve the presentation, we now move the details of their calculations to Supplementary Text 1.

L442-447, Could you please provide a more detailed explanation of the parameter m and how it is set to be species-specific?

Response: We introduced parameter m to account for the saturation effect of HOIs following the theoretical work of Gibbs et al. (2022). We first conducted nonlinear regressions for growth model to estimate species-specific optimum saturating parameter (m) for each species separately. The estimated species-specific m ranged from 0.9×10^{-5} to 2×10^{-3} . We then fitted the growth model again where m was set the same for all species, to make the interaction coefficients comparable among species. The saturating parameter (m) was set within the range from 0.9×10^{-5} to 2×10^{-3} , and one special case $m = 0$. The details are now presented in Supplementary Text 2.1.

L499 Could you please provide a clearer explanation of this equation? While the equation is presented, it would be helpful to offer a detailed explanation of the parameters involved and their meaning.

Response: We have moved all contents related to structural stability into Supplementary Text 3, where we clearly explain the equation of calculating structural stability.

Remarks on code availability

The provided code is clearly structured. However, a key concern is that the most important parts of the analysis—namely, the calculation of interactions and the fitting of the growth and survival models—are demonstrated only using toy data rather than the original dataset. This limitation could affect the reproducibility of the study's main results.

Response: Thank you for raising the concern. We wish we would use the original data to demonstrate the calculations. As noted in the Data Availability statement, the raw census data are only available upon request and with permission of the principal investigators of each forest plot as stated from ForestGEO and CForBio. This is not ideal but is an understandable rule. To improve reproducibility, we have revised the code by replacing the toy data with publicly available census data from the BCI plot. This allows readers to fully reproduce the procedures of estimating interactions coefficients by fitting the growth and survival models with the real BCI data, while ensuring consistency with the data-sharing policies of the contributing plots.

Referee #3 (macroecology):

The authors investigate the latitudinal gradient in species diversity (LDG). A gradient that has been known for a long time but for which the driver(s) remain poorly understood. Conspecific negative density dependence (CNDD) has been mentioned as a potential driver for this gradient but the authors suggest the effect has not been shown conclusively. This is somewhat debatable, as can be seen from references 14 and 15. They show that Higher Order Interactions, interactions of other species on the effect between conspecifics and heterospecifics have a significant effect. The use of HOI is new, for as far as I know. However, in some of the references used, the link between CNDD and the LDG has already been shown, even with data from the same ForestGeo Network. The math is rather complicated and as the text is rather full with mathematical material it is not easy to read. Finally, I missed some discussion on the hypothesis in line 157, and the fact that not only CNDD decreased with latitude but also positive density effects did. Below are some questions and comments that I hope are useful to improve the manuscript.

Hans ter Steege

Response: We hope it is appropriate to say thank you for your insightful feedback that greatly improved our study. As you rightly observed, previous studies have shown links between conspecific negative density dependence (CNDD) and the LDG. However, the relationship is not exclusive but has been much debated over the past decade. Our study shows the limitation of the classical species pairwise CNDD study in inferring the LDG and reveals the importance of high-order interactions in mediating pairwise interactions that in turn affect the LDG. This study reconciles the debate on the role of CNDD in regulating the latitudinal diversity gradient (lines 210-220).

In the revision, we have much improved the presentation of the mathematical details, including using new, more natural notation (e.g. n_{i,i_f} , n_{h,i_f}) to refer neighbor crowding indices and also moving the mathematical details to Supplementary Text 1. Hope these increase the accessibility.

Line 118: This not what ref 11 states, rather the opposite “The proportion of species affected is equivalent to that in tropical forests, failing to support the hypothesis that this mechanism is more prevalent at tropical latitudes”

Response: Thank you for catching this misinterpretation. We replace this reference by two other references (Johnson et al., 2012; LaManna et al., 2017) that support CNDD as a mechanism for the latitudinal diversity gradient (line 136).

Line 121: Here that is even acknowledged.

Line 122. Reference 14, actually shows a rather strong effect of CNDD on the LDG, based on plots of the same ForestGeo network.

Line 122: Ref 15, shows that richer forests in the US show more CNDD, a result also shown here.

Line 122. Hence, I am not so convinced that the findings of the LDG are less inconsistent with the results provided here. Of course, the testing of the HOI’s is new.

Response: True that reference 11 (Hille Ris Lambers et al., 2002, Nature) does not support latitudinal decline of CNDD, at odds with the work of LaManna et al. (2017) and Johnson et al. (2012). We cleaned up the citation in the revision (lines 138-140).

Line 123: add gradient

Response: Added (line 141).

Line 159: I am surprised that in South America the three richest forest plots (Yasuni, Amacayacu, and Manaus) have not been included. La Planada (Colombia) is a high montane forest.

Response: It is highly desirable to include plots from South America. We reached out the PIs in the first place. Unfortunately, we did not receive response. In this round of revision, we reapplied for the access to the plots (and also through personal connections). Again, we were not successful in obtaining the data. That said, we were very pleased to be able to add Rabi plot from Gabon in West Africa to our analysis. We recognize our 32 plots miss sites from South Hemisphere but are pleased that we were able to compile this set of plots, in comparison with other studies, e.g., 21 plots in Wiegand et al. (2025) and 23 plots in Hülsmann et al. (2024), both based on ForestGEO data. To recognize the omission of plots from south of equator (except the Rabi plot), we tune down the “global” scope of our study and reframe our study with a “Northern Hemisphere” focus.

Line 190. But apparently also does positive conspecific facilitation.

Line 191. So why conclude here that only CNDD is a driver of tree diversity, while the effect of CPDD is equally frequent and show a similar latitudinal gradient?

Response: Thanks for raising this point. Previous studies on the latitudinal tree diversity gradient have mainly focused on negative pairwise interactions, i.e., CNDD (e.g., LaManna et al. 2017, Hülsmann et al. 2021, Hülsmann et al. 2024). Our focus on CNDD here is intended to provide comparability with these earlier studies. Indeed, our study found frequent CPDD which also declined with latitude, highlighting the necessity of considering these negative and positive effects simultaneously. To address this, in the revision we assess the relative change in growth rate and survival probability caused by cumulative effects (including both positive and negative effects) of the pairwise interactions and HOIs and how they change with species abundance and latitude. Our results show that latitudinal changes of HOIs enhance the latitudinal gradient, while latitudinal changes of pairwise interactions contribute little to this gradient (lines 222-238).

Line 201. Is that not simply because in higher latitudes there are less species and thus more conspecifics in the surroundings?

Response: Thanks for providing one possible explanation for the observed pattern. As stated in Line 215, higher-order interaction β_{iih} weaken $\alpha_{ii,true}$ more strongly at low latitudes than at high latitudes. The interaction coefficients (both $\alpha_{ii,true}$ and β_{iih}) represent per capita effect (or per capita modification for higher-order coefficients) rather than the cumulative effects (i.e., interaction coefficients times the neighbor crowding index). Therefore, they should not be directly related to the number of conspecific and heterospecific neighbors. We revise the text to clarify the meaning of interaction coefficients and their cumulative effects (lines 422-429).

Line 234. Fig. S2 is the first Suppl Fig called.

Response: In the revision, we have reorganized all the tables and figures in Extended Data and Supplementary Information and carefully checked the order of the citations.

Line 157: Therefore, we hypothesize that higher structural stability in the tropics supports the higher tree diversity – see below

Line 234. This also does not support your hypothesis of line 157. Also it starts to drop of at very low species richness. What does that mean for your hypothesis?

Response: Thank you for noting this ambiguity. Indeed, structural stability is theoretically expected to decline with species richness. That is why we constructed null models and calculated the standardized effect size (SES) of structural stability, thus allowing meaningful comparison across plots with different species richness. Our hypothesis should be more precisely stated as “higher SES of structural stability in the tropics supports higher tree diversity”. To avoid further confusion, we revise the wording accordingly. Following Reviewer #1’s suggestion, we move the entire structural stability analysis to Supplementary Text 3 and add this clarification.

Line 245. But they also strengthen them (Fig 3).

Response: Theoretically speaking, HOIs can either weaken or strengthen pairwise interactions (Figure 1 in Main text). However, our study finds that HOIs on average weaken pairwise interactions in all plots (blue points are closer to zero than orange points in Figure 3 in main text), which is consistent with some previous studies (Mayfield and Stouffer 2017, Li et al. 2021).

L395: How is potential growth determined?

Response: The potential growth rate is the growth rate in the absence of neighbors. It is estimated from the intercept of the growth model. We have now briefly explained that on line 373.

L462: Using the log reduces the errors considerably.

Response: Thank you for raising this point. We agree that log-transforming equation (1) not only linearizes the model but also reduces heteroscedasticity and model residuals, thereby improving the robustness of the estimation. We have added this point in the revision (line 403).

Remarks on code availability:

I am sorry, I did not test the code.

References

Gibbs, T., S. A. Levin, and J. M. Levine. 2022. Coexistence in diverse communities with higher-order interactions. *Proceedings of the National Academy of Sciences* **119**:e2205063119.

Hülsmann, L., R. A. Chisholm, L. Comita, M. D. Visser, M. de Souza Leite, S. Aguilar, K. J. Anderson-Teixeira, N. A. Bourg, W. Y. Brockelman, S. Bunyavejchewin, N. Castaño, C.-H. Chang-Yang, G. B. Chuyong, K. Clay, S. J. Davies, A. Duque, S. Ediriweera, C.

- Ewango, G. S. Gilbert, J. Holík, R. W. Howe, S. P. Hubbell, A. Itoh, D. J. Johnson, D. Kenfack, K. Král, A. J. Larson, J. A. Lutz, J.-R. Makana, Y. Malhi, S. M. McMahon, W. J. McShea, M. Mohamad, M. Nasardin, A. Nathalang, N. Norden, A. A. Oliveira, R. Parmigiani, R. Perez, R. P. Phillips, N. Pongpattananurak, I. F. Sun, M. E. Swanson, S. Tan, D. Thomas, J. Thompson, M. Uriarte, A. T. Wolf, T. L. Yao, J. K. Zimmerman, D. Zuleta, and F. Hartig. 2024. Latitudinal patterns in stabilizing density dependence of forest communities. *Nature*.
- Hülsmann, L., R. A. Chisholm, and F. Hartig. 2021. Is variation in conspecific negative density dependence driving tree diversity patterns at large scales? *Trends in Ecology & Evolution* **36**:151-163.
- Lai, H., Kwek Yan Chong, Alex Thiam Koon Yee, M. M. Mayfield, and D. B. Stouffer. 2022. Non-additive biotic interactions improve predictions of tropical tree growth and impact community size structure. *Ecology* **103**:e03588.
- LaManna, J. A., S. A. Mangan, A. Alonso, N. A. Bourg, W. Y. Brockelman, S. Bunyavejchewin, L.-W. Chang, J.-M. Chiang, G. B. Chuyong, K. Clay, R. Condit, S. Cordell, S. J. Davies, T. J. Furniss, C. P. Giardina, I. A. U. N. Gunatilleke, C. V. S. Gunatilleke, F. He, R. W. Howe, S. P. Hubbell, C.-F. Hsieh, F. M. Inman-Narahari, D. Janík, D. J. Johnson, D. Kenfack, L. Korte, K. Král, A. J. Larson, J. A. Lutz, S. M. McMahon, W. J. McShea, H. R. Memiaghe, A. Nathalang, V. Novotny, P. S. Ong, D. A. Orwig, R. Ostertag, G. G. Parker, R. P. Phillips, L. Sack, I.-F. Sun, J. S. Tello, D. W. Thomas, B. L. Turner, D. M. Vela Díaz, T. Vrška, G. D. Weiblen, A. Wolf, S. Yap, and J. A. Myers. 2017. Plant diversity increases with the strength of negative density dependence at the global scale. *Science* **356**:1389-1392.
- Levine, J. M., J. Bascompte, P. B. Adler, and S. Allesina. 2017. Beyond pairwise mechanisms of species coexistence in complex communities. *Nature* **546**:56-64.
- Li, Y., M. M. Mayfield, B. Wang, J. Xiao, K. Kral, D. Janik, J. Holik, and C. Chu. 2021. Beyond direct neighbourhood effects: higher-order interactions improve modelling and predicting tree survival and growth. *National Science Review* **8**:nwaa244.
- Mayfield, M. M., and D. B. Stouffer. 2017. Higher-order interactions capture unexplained complexity in diverse communities. *Nature Ecology & Evolution* **1**:1-7.

- Wiegand, T., X. Wang, K. J. Anderson-Teixeira, N. A. Bourg, M. Cao, X. Ci, S. J. Davies, Z. Hao, R. W. Howe, W. J. Kress, J. Lian, J. Li, L. Lin, Y. Lin, K. Ma, W. McShea, X. Mi, S.-H. Su, I. F. Sun, A. Wolf, W. Ye, and A. Huth. 2021. Consequences of spatial patterns for coexistence in species-rich plant communities. *Nature Ecology & Evolution* **5**:965-973.
- Wiegand, T., X. Wang, S. M. Fischer, N. J. B. Kraft, N. A. Bourg, W. Y. Brockelman, G. Cao, M. Cao, W. Chanthorn, C. Chu, S. Davies, S. Ediriweera, C. V. S. Gunatilleke, I. A. U. N. Gunatilleke, Z. Hao, R. Howe, M. Jiang, G. Jin, W. J. Kress, B. Li, J. Lian, L. Lin, F. Liu, K. Ma, W. McShea, X. Mi, J. A. Myers, A. Nathalang, D. A. Orwig, G. Shen, S.-H. Su, I. F. Sun, X. Wang, A. Wolf, E. Yan, W. Ye, Y. Zhu, and A. Huth. 2025. Latitudinal scaling of aggregation with abundance and coexistence in forests. *Nature* **640**:967-973.

Dear Dr. [REDACTED],

Thank you for reviewers' positive feedback and your interest in considering our work. Again, we have very carefully addressed all the remaining concerns and suggestions of the reviewers. Changes have been made in the main text, extended data and supporting information (see our responses to reviewers' comments on the pages following this letter). In addition, we have thoroughly proofread the entire manuscript and corrected grammatical and stylistic issues to improve the clarity, interpretability, and overall presentation of the study. All responses to the comments and changes of the manuscript are marked in blue.

Sincerely,

Chengjin Chu and co-authors

Referee #1

The new version of the manuscript improved substantially and the authors have done a great job of answering my questions and implementing changes! Overall, the study shows high originality and significance and is a long-needed contribution that may change the way species interactions in plant communities are conceptualized. The data are impressive and the methodology is valid, but needs a little more explication. The revision now has a clearer storyline and the methods gained substantially in clarity, as the text and the methods are now focusses on the heart of the analysis, with the robustness tests, structural stability, and the complicated-looking equations for the detailed calculations of the different crowding indices moved into the supplement. The new notation of the equations also helps. Nevertheless, I have a number of more minor questions and suggestions to strengthen the line of arguments of the manuscript. Additionally, a grammar check may be needed as there are several somewhat unusual phrases.

Response: We are grateful for your insightful evaluation on the originality, significance, and potential conceptual impact of our study and for the further comments and suggestions which greatly improved the presentation and clarity of our study. In the following, we explain how we incorporate and address your comments/suggestions in our revision. In addition, we have thoroughly proofread the entire manuscript and corrected the grammatical and stylistic issues as much as we can.

General comments

1) Thank you for re-running the analysis for large and small trees separately, it is good that the results for large trees are consistent with the results shown in the main text for all trees. You may include the analogous analysis to figure 5 for large trees to the supplement.

Response: Thanks for the helpful suggestion. We have conducted separate analyses of the relative changes in growth rate for small and large trees, and added corresponding results in Supplementary Information (lines 107-117, Tables S7 and S8, Figs. S7 and S8).

For both size classes, the relative changes in growth rate caused by cumulative effects of pairwise interactions and HOIs both shift from beneficial for rare species to detrimental for common species (Figs. S7 and S8). The stabilizing effect of pairwise interactions remains relatively constant across latitudes for both small and large trees (Tables S7 and S8). By contrast, the stabilizing effect of HOIs becomes weaker towards higher latitudes for small trees (Table S7) but remains relative constant for large trees (Table S8). These patterns are also consistent with results for all trees combined, except that the latitudinal decline in stabilizing effect of HOIs for large trees is not significant ($P = 0.665$, Table S8), probably reflecting the smaller sample size of large trees.

2) I am happy that you now examine the question (Q3) of how the cumulative effects of pairwise and higher-order interactions on growth (and survival) change with species abundance and latitude (instead of conducting A structural stability analysis). I found the new figure 5 that reports these results very interesting (i.e., a rare species advantage with respect to growth, which was stronger at lower latitudes).

Response: We are pleased that the revised approach to addressing Q3, focusing on how the relative changes in growth (and survival) caused by cumulative effects of pairwise interactions and HOIs vary with species abundance and latitude, is considered appropriate. We also appreciate your scrutiny of the new Fig. 5 and the interpretation of the rare-species growth advantage and its stronger expression at lower latitudes. Thank you!

3) Thank you for showing the correlations between the different pairwise and higher-order neighbourhood crowding indices. This figure should go into the supplement to allow the reader a better understanding of the properties of the higher-order indices. You may also mention that the correlation between $n_{i,if}$ and $n_{ii,if}$ arises because $n_{ii,if}$ is approximately the square of $n_{i,if}$, and the correlation between the heterospecific indices $n_{i,if}$ and $n_{hh,if}$ arise because $n_{hh,if}$ is approximately the square of $n_{h,if}$.

Ok, your argument is that you can retain both the pairwise and the HOI terms because this would be similar to using a linear term and a squared term in a regression. This should also be mentioned in the methods.

However, I would propose a somewhat different, more biological argument for retaining pairwise and HOI terms. First, comparing e.g., equations S3 and S8, you see that the higher-order crowding index $n_{ih,if}$ weights each conspecific neighbour i_p by (i) its DBH, (ii) its distance $d[if,ip]$ to the focal individual if , and (iii) by its heterospecific crowding index $n_{h,ip}$. Thus, the indices $n_{ih,if}$ and $n_{ii,if}$ refine the pairwise index $n_{i,if}$ in the same way as weighting by DBH refines the crowding index that just counts neighbours and weights them by distance.

Analogously, the indices $n_{hh,if}$ and $n_{hi,if}$ refine the index $n_{h,if}$.

The argument of weighting (e.g., in case of competitive interactions) by $1/d[if,ip]$ is that more distant neighbours compete less, and the argument of weighting by its DBH is that larger trees may compete more strongly. Now, this argumentation does not really work for the weighting with $n_{h,ip}$ because a conspecific neighbour i_p with a very large index $n_{h,ip}$ should show a weaker competitive ability, but not a larger one as suggested by the over-proportionally large weight $n_{i,ip}$. However, weighting by $(1 - \alpha n_{h,ip}/E[n_{h,ip}])$, instead of weighting by $n_{h,ip}$, does the job, which basically introduces the pairwise term! $E[n_{h,ip}]$ is the expectation of $n_{h,ip}$ for conspecifics i_p , and α is the strength of the higher-order effect. The term $\alpha/E[n_{h,ip}]$ is implicitly included in your regression coefficients α_{ii} and β_{iih} , but it might be interesting to calculate α . The weight $(1 - \alpha n_{h,ip}/E[n_{h,ip}])$ explains also the negative correlation between α_{ii} and β_{iih} and between α_{ih} and β_{ihh} .

Response: Thank you for this detailed and constructive comment, and for providing alternative biological interpretation of retaining both pairwise and HOI terms, as well as of the correlations between α_{ii} and β_{iii} (and between α_{ih} and β_{ihh}).

Following your suggestion, we have moved the figure showing correlations among the pairwise and higher-order crowding indices to the Supplementary Information (Fig. S2) and explicitly

stated the strong correlations between n_{i,i_f} and n_{ii,i_f} , and between n_{h,i_f} and n_{hh,i_f} (lines 43-46 in Supplementary Information).

To clarify the biological interpretation of retaining pairwise and HOI terms, we illustrate it using the Lotka-Volterra model (without considering individual size and distance for simplicity). In the absence of HOIs, the classic Lotka-Volterra model is written as:

$$\frac{1}{N_i} \frac{dN_i}{dt} = r_i + \alpha_{ii}N_i + \alpha_{ih}N_h \quad (1)$$

where α_{ii} and α_{ih} are constants representing intraspecific and interspecific interaction coefficients, respectively. When the four types of HOIs (i.e., (i) α_{ii} modified by conspecifics ($\beta_{iii}N_i$), (ii) α_{ii} modified by heterospecifics ($\beta_{iih}N_h$), (iii) α_{ih} modified by conspecifics ($\beta_{ihh}N_h$), (iv) α_{ih} modified by heterospecifics ($\beta_{ihh}N_h$)) are included, equation 1 becomes:

$$\begin{aligned} \frac{1}{N_i} \frac{dN_i}{dt} &= r_i + (\alpha_{ii} + \beta_{iii}N_i + \beta_{iih}N_h)N_i + (\alpha_{ih} + \beta_{ihh}N_h)N_h \\ &= r_i + \alpha_{ii}N_i + \alpha_{ih}N_h + \beta_{iii}N_i^2 + \beta_{iih}N_iN_h + \beta_{ihh}N_h^2 \quad (2) \end{aligned}$$

The β implies the **absolute modification** of α . This formulation is consistent with a broad range of previous theoretical (Bailey et al. 2016, Letten and Stouffer 2019, Gibbs et al. 2022) and empirical studies (Mayfield and Stouffer 2017, Buche et al. 2024).

We appreciate your suggestion to consider an alternative formulation in which HOIs act as **relative modifiers** of pairwise coefficients:

$$\begin{aligned} \frac{1}{N_i} \frac{dN_i}{dt} &= r_i + \alpha_{ii}(1 + \beta'_{iii}N_i + \beta'_{iih}N_h)N_i + \alpha_{ih}(1 + \beta'_{ihh}N_h)N_h \\ &= r_i + \alpha_{ii}N_i + \alpha_{ih}N_h + \alpha_{ii}\beta'_{iii}N_i^2 + \alpha_{ii}\beta'_{iih}N_iN_h + \alpha_{ih}\beta'_{ihh}N_h^2 \\ &\quad + \alpha_{ih}\beta'_{ihh}N_h^2 \quad (3) \end{aligned}$$

Simple relationships between the two parameterizations are obtained:

$$\beta_{iii} = \alpha_{ii}\beta'_{iii}, \beta_{iih} = \alpha_{ii}\beta'_{iih}, \beta_{ihh} = \alpha_{ih}\beta'_{ihh} \quad (4)$$

Your alternative formulation provides an intuitive biological perspective in which HOIs scale the strength of pairwise interactions, and a natural explanation for the correlations between α and β .

Because both formulations are mathematically equivalent under reparameterization and because the absolute-modification form in equation (2) is standard in the HOI literature, we retain this representation in the present study. We also explicitly state exploration of alternative formulations as an important future research direction in the main text (lines 256–261).

Finally, regarding your argument that “a conspecific neighbor i_p with a large index n_{h,i_p} should show a weaker competitive ability”, our current interpretation is that this depends on the sign of the modification. When heterospecific neighbors weaken intraspecific pairwise interactions α_{ii} (Fig. 1c), larger n_{h,i_p} indeed implies weaker competitive effects. However, when heterospecific neighbors strengthen α_{ii} (Fig. 1b), larger n_{h,i_p} would instead imply stronger competitive effects.

We therefore do not impose a fixed weighting of the form $\left(1 - \alpha \cdot \frac{n_{h,i_p}}{E(n_{h,i_p})}\right)$ in the present model.

Bairey, E., E. D. Kelsic, and R. Kishony. 2016. High-order species interactions shape ecosystem diversity. *Nature Communications* **7**:1-7.

Buche, L., I. Bartomeus, and O. Godoy. 2024. Multitrophic higher-order interactions modulate species persistence. *The American Naturalist* **203**:458-472.

Gibbs, T., S. A. Levin, and J. M. Levine. 2022. Coexistence in diverse communities with higher-order interactions. *Proceedings of the National Academy of Sciences* **119**:e2205063119.

Letten, A. D., and D. B. Stouffer. 2019. The mechanistic basis for higher-order interactions and non-additivity in competitive communities. *Ecology Letters* **22**:423-436.

Mayfield, M. M., and D. B. Stouffer. 2017. Higher-order interactions capture unexplained complexity in diverse communities. *Nature Ecology & Evolution* **1**:1-7.

Specific comments

line 115-117

Perhaps better “Theory suggests that (...) may greatly modify the pairwise interactions”

Response: Revised (lines 116-118).

line 118-119

Perhaps: “However, there is a lack of empirical studies that investigate how HOIs intertwine with pairwise interactions, and how they may contribute to the latitudinal diversity gradient.”

Response: Revised (lines 118-119).

line 122-124

There is still a gap between the rare species advantage with respect to growth and survival and promotion of species diversity. So I suggest to be more conservative here, for example: “More importantly, HOIs were found to benefit rare species while disadvantaging common species, which suggests a mechanism to promote species diversity.”

Response: We agree that it is more appropriate to be more conservative about concluding HOIs as a mechanism promoting species diversity from rare species advantage with respect to growth and survival, and we have revised the text accordingly (line 124).

line 148

At this stage of the manuscript, you propose the hypothesis that HOIs may have an important effect, I would therefore suggest to add “potential effects” to make this clearer: “(...) neglecting the potential effects of higher-order interactions (HOIs) that (...)”

Response: Added (line 149).

line 187

I would add: “on growth and survival”: “(...) of pairwise interactions and HOIs on growth and survival (...)”

Response: Added (line 189).

lines 193-194

What's about tree survival rates? Add the panel analogous to Fig. 3b to Extended Data Figure 3.

Response: HOI-inclusive models only slightly improve predictions of survival probability of trees compared to the alternative models. We have added this result to Extended Data Figure 3b.

Extended Data Figures 1, 4

lines 11, 40, 41

There are typos in the subscripts, it must be: “(...) higher-order interaction coefficients β_{ih} and β_{ih} .”

Response: Thanks for pointing out the typos. Corrected (lines 11, 46 in Extended Data Tables and Figures).

lines 202-206, Figure 4

The result that some α and β values decrease with increasing latitude is an interesting pattern that apparently suggests that the average strength of true intraspecific pairwise interactions ($\alpha_{ii,true}$) and of HOIs would decline rapidly with latitude. Because of this catchy suggestion, it is even more important to ensure that this pattern is not the result of a systematic change in another variable across latitude.

Your scaling of the different indices $n_{i,if}$, $n_{ii,if}$, $n_{ih,if}$, and $n_{hi,if}$ with conspecific abundance N_i (equations S17 and S18) suggests that this result could potentially be caused by systematic changes in the values of the crowding indices with latitude. Note that $n_{h,if}$ and $n_{hh,if}$, which scale with heterospecific abundance N_h (that varies little compared to N_i), do not show the declining trend.

For example, the impact of the conspecific pairwise interactions on growth is $\exp(\alpha_{ii}n_{i,i_f})$ and n_{i,i_f} scales with conspecific abundance N_i (eq. S17). Given that there are fewer species in forests at higher latitudes, species abundances will tend to be higher and therefore n_{i,i_f} will also tend to be higher. It is therefore possible that systematic changes in n_{i,i_f} with latitude may counteract the trend in α and β values and lead across latitude to similar range of the net effects of pairwise interactions on growth [i.e., $\exp(\alpha_{ii}n_{i,i_f})$]. To exclude this possibility and to verify that the net effect of conspecific interactions on growth (and perhaps survival) decline with latitude, you could expand Figure 5 to show the relative changes in growth rate along latitude due to the six different components $\exp(\alpha_{ii}n_{i,i_f})$, $\exp(\alpha_{ih}n_{h,i_f})$, $\exp(\beta_{iii}n_{ii,i_f})$, $\exp(\alpha_{iih}n_{ih,i_f})$, etc. for two or three typical abundances.

Response: Thanks very much for your deep thinking on this point. We fully agree that it is essential to ensure that the observed latitudinal declines in interaction coefficients are not artefacts arising from systematic changes in the crowding indices with latitude.

Following your suggestion, we have examined the latitudinal change in crowding indices (n_{i,i_f} , n_{h,i_f} , n_{ii,i_f} , n_{ih,i_f} , n_{hi,i_f} and n_{hh,i_f} , Fig. R1), and cumulative effects ($\exp(\alpha_{ii}n_{i,i_f})$, $\exp(\alpha_{ih}n_{h,i_f})$, $\exp(\beta_{iii}n_{ii,i_f})$, $\exp(\beta_{iih}n_{ih,i_f})$, $\exp(\beta_{ihh}n_{hh,i_f})$, Fig. R2) of pairwise and higher-order interactions. As you expected, the pairwise conspecific crowding index n_{i,i_f} does increase with latitude (Fig. R1a), which counteracts the decreasing trend in α_{ii} (Fig. 4a in main text) and leads to a slight increase in $\exp(\alpha_{ii}n_{i,i_f})$ with latitude (Fig. R2a).

Despite of this, we do not think that the latitudinal decline observed for α_{ii} is a result of systematic changes in n_{i,i_f} . In the Lotka-Volterra model, the species abundance at equilibrium is determined by intrinsic growth rates and per capita interaction coefficients. Therefore, the crowding indices (similar to equilibrium abundance) are the outcome rather than the cause of

species interactions. Consistent with this view, we find that a latitudinal decrease in n_{h,i_f} (Fig. R1b) is not associated with a corresponding increase in α_{ih} (Fig. 4b in main text).

Finally, the latitudinal biotic interaction hypothesis, including CNDD, is typically formulated in terms of interaction intensities (coefficients) rather than the cumulative effects. For consistency, we retained Fig. 4 in the main text reporting the latitudinal change in coefficients of biotic interactions.

Figure R1. Latitudinal changes in pairwise and higher-order crowding indices.

Figure R2. Latitudinal changes in cumulative effects of pairwise and higher-order interactions.

line 213

add: “than by conspecific neighbors.”

Response: Added (line 215).

lines 226-238

I think this is a good idea to show in the main text how the pairwise and higher-order interactions modify the growth of individual trees on average across latitude to provide an answer to question Q3. The results are very interesting!

From lines 423-427 it follows that values of RCPAIRS and RCHOIs larger than one promotes the growth on average (and values smaller than one reduces the growth on average). To facilitate interpretation of Figure 5, I would keep the logarithmic y-axis in Figure 5, but show instead of the log-transformed axis values (i.e., 4, 2, 0, -2 and -4) the corresponding untransformed values. With this small change, the reader can easily assess the magnitude of the average impact of the different interaction types.

For extended Data Figure 5 the interpretation becomes more complicated because survival does not scale linearly with the terms $\exp(\text{pair})$ and $\exp(\text{HOI})$. How did you calculate the relative changes in survival due to pairwise and HOI interactions?

Response: Thanks for this great suggestion. We have now modified the y-axis of Fig. 5 and Extended Data Fig. 5 to display the corresponding untransformed values while retaining the logarithmic scale, which makes it much easier to interpret the magnitude of the effects.

Yes, the calculation of relative changes differs slightly between growth and survival. The growth model can be linearized by log-transforming Equation (1), while the survival model can be linearized by logit-transforming $\left[\log\left(\frac{p}{1-p}\right)\right]$ Equation (2). Consequently, the cumulative effects of pairwise interactions ($e^{PAIR_{if}}$) and HOIs ($e^{HOI_{if}}$) for survival actually quantify the odds $\frac{p}{1-p}$ rather than directly in the survival probability p .

Because the odds $\frac{p}{1-p}$ is a positive and monotonic function of p , variation in survival probability with species abundance and latitude exhibits the same qualitative patterns as variation in the odds $\frac{p}{1-p}$ (compare Table R1 and Fig. R3 with Extended Data Table 2 and Extended Data Fig. 5). To avoid unnecessary methodological complexity in the main text, we have added this explanation to the legend of Extended Data Table 2 and Extended Data Fig. 5.

Table R1 Summary of relative changes in survival probability varying with species abundance and absolute latitude.

Response	Effect	Estimate	Standard error	t value	P value
RC _{PAIR}	Intercept	0.389	0.430	0.907	0.365
	Abundance	-0.199	0.128	-1.558	0.120
	Latitude	-0.026	0.020	-1.333	0.183
	Abundance:Latitude	0.006	0.005	1.195	0.232
RC _{HOI}	Intercept	21.778	2.641	8.245	3.91×10 ⁻¹⁶
	Abundance	-4.527	0.785	-5.769	9.88×10 ⁻⁹
	Latitude	-0.314	0.122	-2.573	0.010
	Abundance:Latitude	0.076	0.032	2.362	0.018

Figure R3. Predicted relationships between relative change in survival probability and species abundance across three latitudinal geographic zones.

lines 233-236

Not really, the effect of pairwise interactions is weaker (with the slope becoming non-significant for temperate forests), but not “relative constant”. Using a linear scaling of the y-axis (i.e., the untransformed RCPAIRS and RCHOIs values) would show this clearer.

lines 236-238

Perhaps: “Taking together, our findings suggest that latitudinal changes in HOIs and pairwise interactions may strengthen the latitudinal tree diversity gradient by promoting growth of less abundant species at lower latitudes more strongly, with weaker contributions from pairwise interactions.”

Response: Thanks for this careful reading. We agree that Fig. 5 in main text visually suggests the stabilizing effect of pairwise interactions becomes weakened towards higher latitudes. However, this trend is not statistically supported: the interaction between abundance and latitude for RC_{PAIR} is not significant ($P = 0.736$ in Table 1 in main text). For this reason, we revised “relatively constant” to “changes little” (line 237).

lines 422-424

To further assess how the latitudinal change of pairwise interactions and HOIs may contribute to the latitudinal tree diversity gradient (Q3), we first calculated the relative change in growth rate and survival probability caused by the neighbourhood (...)

Response: Thanks for editing this sentence. Revised (lines 439-440).

Supplementary Text

lines 70-133 are missing.

Response: Thanks for pointing out this issue. The missing lines were due to a display error in the uploaded file. We will carefully check the supplementary file to ensure that all lines are properly displayed when we resubmit.

Referee #2

The authors have done a great job addressing the previous concerns and comments raised. I only have some minor comments now.

Response: We thank you for this very positive assessment and are pleased that the previous concerns and comments have been fully addressed. In this revision, we have further carefully addressed your additional comments, as detailed below.

L138-139 Part of the statement is wrong. Ref 11 (Daniel J. Johnson 2012), they found “species-rich regions exhibited stronger CNDD than species-poor regions” instead of “significant decline in CNDD with increasing latitude or species richness”. Please revise it.

Response: Thanks for pointing out the difference. Revised (line 140).

L227 Does “cumulative effects” refer to the combined influence of both competitive and facilitative effects? If so, please clarify this explicitly when the term is first introduced.

Response: Yes, the “cumulative effects” included both competitive and facilitative effects. We have added this clarification (line 230).

L377 what is λ_i ? Does it have any ecological meaning?

Response: λ_i represents the intrinsic survival probability of species i , similar to G_i in growth model. We added this interpretation (line 395).

I wonder whether, in the scenario β_{ihh} , you also consider cases in which the heterospecific neighbour is a third species, rather than one of the two species involved in the focal pairwise interaction. My understanding is that this is the case; however, the notation “ β_{ihh} ” suggests otherwise and could be potentially confusing.

Response: Yes, β_{ihh} includes the case in which the heterospecific neighbor is a third species. However, it is not possible to measure all species-specific pairwise and higher-order interaction coefficients in species-rich forests. Instead, we group neighbors into “conspecifics” and “heterospecifics”. Therefore, β_{ihh} indicates interspecific pairwise interactions modified by heterospecific neighbors (as a general term), which includes the case where neighbor and neighbor’s neighbor are the same species ($\beta_{ijj}: \alpha_{ij} \leftarrow j$) and the case where neighbor and neighbor’s neighbor are different species ($\beta_{ijk}: \alpha_{ij} \leftarrow k$).

Supplementary text L70-134, please check, the content is incomplete.

Response: Thanks for pointing out this issue. The missing lines were due to a display error in the uploaded file. We will carefully check the Supplementary file to ensure that all lines are properly displayed when we resubmit.

Referee #3

I would like to thank the authors for a thorough rewrite. Although the math is still complicated to a general community ecologist like myself, I am convinced that the analyses and the text are strong. Just a few things caught my eye when reading

Response: We sincerely thank you for this encouraging assessment and for taking the time to carefully evaluate the revised manuscript. We also appreciate the additional points noted below and have addressed each of them in detail in the following responses.

line 234: should read 'relatively constant'

This phrase was revised as “changes little” for more accurate description (line 237).

line 240-241: I would suggest "We would like to note some limitations of our analyses for future consideration, however." Otherwise, there is a focus on 'we'.

We revised this sentence for clarity (line 245).

line 246: 'upscaled to the communities'

Corrected (line 251).

with kind regards

Hans ter Steege

I congratulate the authors on their thorough revision of the manuscript and for carrying out all the additional analyses in response to my comments, which have clarified my remaining queries. The manuscript is an important contribution!

I only have a few minor comments left. I am fine with the absolute-modification form in equation (2) for interpreting the pairwise and HOI terms. However, it is important to briefly mention in the methods how the pairwise and HOI crowding indices are estimated (see below).

Response: Thank you for the careful evaluation of our revised manuscript and for the constructive comments provided in the previous rounds of review, which have greatly improved the clarity and rigor of our manuscript. We appreciate your positive assessment of our work and are pleased that our additional analyses and revisions have addressed your previous concerns. Below, we respond to your new comments.

Specific comments

lines 120-122

To be more specific, you may add that you analyze survival and growth of individual trees.

Response: Done as suggested (lines 125-126).

lines 149-150

This is true for the Lotka-Volterra type models that operate at the species level. However, in your study you consider survival and growth of individual trees based on neighborhood interactions. I would therefore suggest to change in according to figure 1 to “the potential effects of higher-order interactions (HOIs) that emerge when pairwise interactions between two neighbored trees are modified by other neighbors (Fig. 1).”

Response: Done as suggested (line 154).

line 185-186

For better understanding add “, the true interaction between two trees of species i and j in the absence of other neighbors”:

(...) whereas pairwise interactions estimated from HOI-inclusive models isolate HOIs and thus represent $\alpha_{ij,true}$ (orange arrows in Fig. 1), the true interaction between two trees of species i and j in the absence of other neighbors.

Response: Done as suggested (line 191).

Figure 5

It would be interesting to see also the relative change in growth rate caused by the cumulative effects of the modified pairwise interactions (i.e., based on the PAIR-only model).

Response: Thanks for this interesting suggestion. Following this comment, we examined the relationship between relative change in growth rate caused by cumulative effects of modified pairwise interactions and species abundance, and how this relationship changes with latitude (Fig. R1).

Conceptually, the modified pairwise interactions ($\alpha_{ij,modified}$) represent the net outcome of true pairwise interactions ($\alpha_{ij,true}$) and higher-order interactions (β_{ijk}) (Fig. 1 in main text).

However, they are estimated independently from different models: $\alpha_{ij,modified}$ from PAIR-only model, $\alpha_{ij,true}$ and β_{ijk} from HOI-inclusive model. Therefore, no simple relationship such as $\alpha_{ij,modified} = \alpha_{ij,true} + \beta_{ijk}$ can be derived between them. Consequently, the patterns derived from the PAIR-only model do not directly correspond to those obtained when true pairwise and higher-order effects are explicitly separated in HOI-inclusive model. The discrepancy between Fig. R1 (modified pairwise interactions) and Fig. 5 (true pairwise and higher-order interactions) highlights the importance of explicitly separating these components, as lumping them into modified pairwise interactions may lead to different or even opposing results. We aim to disentangle the contributions of true pairwise interactions and higher-order interactions, and thus retain the results from HOI-inclusive models in the main text.

Figure R1 | Predicted relationships between relative change in growth rate caused by cumulative effects of the modified pairwise interactions and species abundance across three latitudinal geographic zones. Predictions are shown for three geographic zones: tropical (0° – 23.5° , a), subtropical (23.5° – 35° , b), and temperate (35° – 66.5° , c).

Line 241

You could start a new paragraph with “Taken together”.

Response: We actually started a new paragraph with “Our study ...”, however, we forgot to the space at beginning of paragraph, which makes it like one paragraph with previous content. We have corrected this issue (line 249).

line 412

It is important to give the reader an idea of how the crowding indices and the higher order crowding indices are estimated. You may add before “The calculation (...)”:

“In short, the conspecific and heterospecific pairwise neighborhood crowding indices n_{i,i_f} and n_{h,i_f} add up the crowding contributions of all conspecific and heterospecific neighbors of a focal individual within a given radius, respectively. The contribution of a neighbor is an increasing function of the dbh of the neighbor and a decreasing function of the distance to the focal individual (eq. S3, S4). The contribution of a neighbor in the corresponding higher-order crowding indices n_{ii,i_f} and n_{hi,i_f} is additionally multiplied by its conspecific crowding index (eq.

S7, S9), and that of n_{ih,i_f} and n_{hh,i_f} is additionally multiplied by its heterospecific crowding index (equations S8, S10).”

Response: Thanks for this helpful suggestion. Done as suggested (lines 472-479).

line 441

add “true”: “true pairwise interactions”

Response: Added (line 508).